# CEDAR: Agent-Orchestrated Tree Search for Goal-Directed Optimization of Complex Systems

## Abstract

Complex systems modeling analyzes nonlinear, feedback-driven phenomena from population dynamics to economic policy, supporting decisions with significant societal impact. In established practice, models are often authored in specialized system-dynamics languages (e.g., DYNAMO, STELLA) that specify the models' structure. However, building and refining such models requires extensive manual effort due to (1) the opaque relationship between the structure and emergent behavior and (2) the labor-intensive workflows imposed by these languages. These barriers limit adoption and hinder effective decision-making. To address these challenges, we introduce CEDAR (Complex-systems Exploration and Design via Agent-Orchestrated Refinement), an autonomous method that uses LLM (Large Language Model) agents to discover and improve complex systems that satisfy user-specified goals. Our key innovation is an LLM-driven MCTS (Monte Carlo Tree Search) process deeply coupled with complex system: at each iteration, an LLM Judge evaluates performance against goals and an LLM Editor proposes improved system variants. We represent systems using a restricted, runnable subset of Python with domain-specific primitives, enabling LLMs to meaningfully and modify system dynamics directly. CEDAR is theoretically designed with formalization in mind, and empirically enables automatic optimization of vague goals, thereby reducing human effort while achieving capabilities beyond existing approaches. The unified design handles diverse systems across domains, constructing complex systems that would otherwise require extensive manual fitting. Moreover, by using LLMs to interpret systems, CEDAR makes system design transparent and accessible, facilitating broader adoption of complex systems modeling.

## 1 Introduction

Computational system modeling has been essential for analyzing complex systems across diverse domains, from global dynamics (Forrester, 1971; 2011) and pandemic diffusion (Zhu et al., 2025) to social simulations (Kolson, 1996) and biological systems (Ruth & Hannon, 2012; Hannon & Ruth, 2014). These approaches, supported by specialized languages and tools (Radzicki & Taylor, 2011; Richmond, 1985), formalize feedback loops and nonlinear interactions to explore system behavior beyond simple simulation and forecasting. Such systems thinking (Anderson & Johnson, 1997; Arnold & Wade, 2015), by attempting to model the subject as a system, critically supports policy design and strategic decision-making (Peterson & West Lebanon, 2003).

However, a central challenge remains: extensive human effort. This effort stems from the opaque relationship between model structures and emergent behavior, combined with the manual workflows imposed by established modeling languages. These factors limit adoption and hinder timely decision-making across domains where complex systems modeling could provide valuable insights.

Large language models (LLMs) and LLM-powered agents now offer unprecedented capabilities in reasoning and planning for complex systems. Recent work, such as AI Scientist (Yamada et al., 2025) and related efforts (Kumar et al., 2025), demonstrates automated discovery in scientific and artificial-life domains. These advances suggest that we can now automate the discovery and optimization of complex systems themselves: leveraging cognitive agents to autonomously improve system structures.

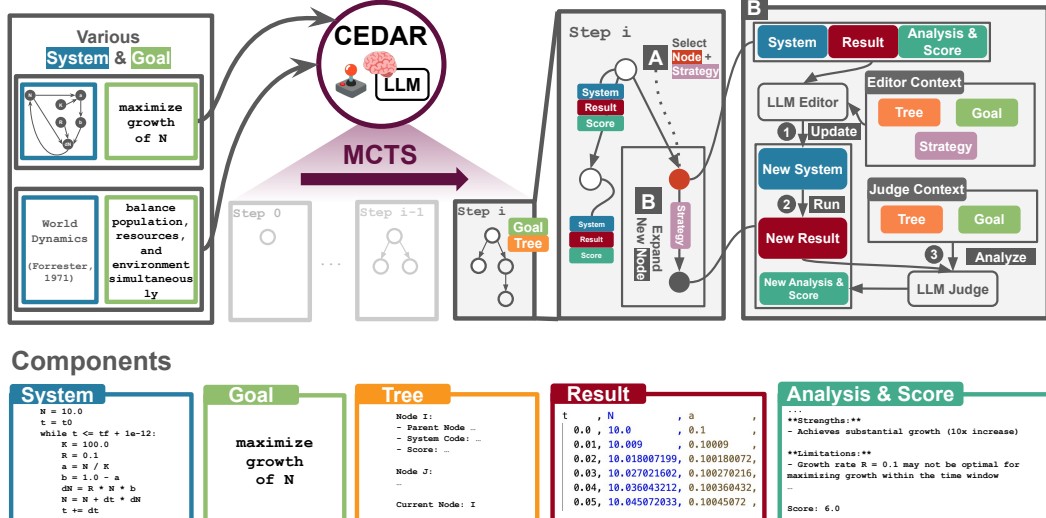

Figure 1: Overall architecture of CEDAR. The autonomous approach uses Monte Carlo Tree Search (MCTS) to iteratively select system variants for expansion. At each iteration, an LLM Editor generates improved system variants and executes them, then an LLM Judge evaluates their performance and provides detailed analysis. The method integrates several key components: a system $P$ represented by a restricted subset of Python code, a natural-language goal description, the search tree structure, system execution results, and comprehensive analysis feedback.

To address these challenges, we propose an autonomous method that combines Monte Carlo Tree Search (MCTS), large language models, and a unified system representation to discover complex systems that meet specified goals. At each iteration, MCTS selects a system variant for expansion, where an LLM Editor generates improved system variants, and the system is executed and evaluated by an LLM Judge for performance and analysis. This process is enabled by our unified representation using a restricted subset of Python with domain-specific mathematical primitives and extensive inline documentation, thereby leveraging Python's familiarity and LLMs' strong Python capabilities for seamless integration between search, evaluation, and modification components.

Our method addresses key limitations in system modeling through four main contributions. First, it enables automatic optimization for vague goals, reducing human effort while achieving capabilities beyond existing approaches. Second, the unified design accommodates diverse systems, demonstrating generality across domains. Third, it constructs complex systems that would otherwise require extensive manual fitting, improving efficiency. Finally, by leveraging LLMs to interpret systems, it makes system design transparent and accessible to a broader range of users.

## 2 RELATED WORKS

The central motivation of our work is to modernize **(1) the modeling of complex systems** using **(2) tree search algorithms with LLMs** to **(3) optimize systems in an LLM-driven way**. In this section, we discuss these three lines of work that are related to our motivation.

**(1) Complex Systems and How to Model Them**   *World Dynamics* (Forrester, 1971) introduced a computational framework for modeling the planet as a nonlinear, feedback-driven system. Using the DYNAMO language for system dynamics analysis (Radzicki & Taylor, 2011), it simulated population, resources, and industrial output over long horizons, marking a paradigm shift from linear prediction to dynamic system-level exploration (Forrester, 2011). Since then, complex systems have been applied to pandemic diffusion (Zhu et al., 2025), war games, and social simulations (Kolson, 1996). Visual programming languages like STELLA (Richmond, 1985) were developed as successors to DYNAMO, with applications in economics, thermodynamics, and biological systems (Ruth & Hannon, 2012; Hannon & Ruth, 2014). Examples of DYNAMO and STELLA can be found online.

Computational system modeling has been essential for analyzing complex systems and their dynamics. Computation can be used not only to simulate and forecast but also to explore system behavior by formalizing feedback loops and nonlinear interactions. Such systems thinking (Anderson & Johnson, 1997; Arnold & Wade, 2015) supports policy design and strategic decision-making (Peterson & West Lebanon, 2003).

Despite these advances, significant challenges remain. Traditional system dynamics relies on expert-defined variables and relationships (Ford & Sterman, 1997; Bérard, 2010; Zagonel, 2002; Vennix et al., 2021). The main challenge is the extensive human effort required. First, the relationship between model structures and emergent system behavior is often unclear (Schoenberg, 2019; Güneralp, 2004; Barlas, 1996). Second, established modeling languages require manual, labor-intensive workflows that can be counterintuitive (MIT System Dynamics in Education Project, 1998; Hines, 1996; Sterman, 2000). The central motivation for our work is to modernize complex system design and iteration workflows traditionally centered on manual modeling.

**(2) Tree Search Algorithms with LLMs** Monte Carlo Tree Search (MCTS) (Kocsis & Szepesvári, 2006) has been successfully applied to various optimization problems, providing a foundation for CEDAR. Combining MCTS with neural networks has led to superhuman performance in games (Silver et al., 2016; 2017b;a; Schrittwieser et al., 2020) and to the discovery of novel algorithms (Fawzi et al., 2022; Mankowitz et al., 2023) and neural network architectures (Nasir et al., 2024).

More recently, large language models (LLMs) have been combined with MCTS to enhance its capabilities. Generic approaches that improve MCTS involve LLM-powered value functions and self-reflection (Zhou et al., 2024), as well as LLM-powered node selection and expansion strategies (Inoue et al., 2025). Additional works apply MCTS with LLMs to specific tasks, such as automatic design (Zhang et al., 2025b) and agent collaboration (Gan et al., 2025).

Notably, MCTS with LLMs offers strong capabilities in two areas: planning and reasoning. For planning, recent works propose accelerating planning by letting LLMs serve as common-sense policy priors (Zhao et al., 2023), improving planning with transformer-based search dynamics (Lehnert et al., 2024), and enhancing interpretability through contrastive MCTS reasoning (Gao et al., 2024). For reasoning, LLMs have been used to guide MCTS traversals of knowledge graphs (Song et al., 2025) and to refine reasoning through intermediate steps (Chi et al., 2025).

**(3) LLM-based Optimization for Systems** LLMs with MCTS show promising results for complex search. AI Scientist (Yamada et al., 2025) enables automated scientific discovery, Kumar et al. (2025) extend this to artificial life simulations, and Katz et al. (2024) optimize runnable code for reasoning. Recent work applies LLMs to system-dynamics modeling (Liu et al., 2024; Luo et al., 2025; Liu & Keith, 2025), treating models as a predictors of behavior that are *unknown* to the optimizer, whereas we directly optimize *known* system models for goal-directed outcomes. These studies consider small systems (up to 4 variables and 12 steps), while our method scales to roughly 4000 steps and dozens of variables, making direct extensions of their setups non-trivial.

**(4) Prior works close to our work.** Adjacent works combine LLMs with search for different goals. I-MCTS (Liang et al., 2025) targets AutoML hyperparameters, ChemReasoner (Sprueill et al., 2024) and CheMatAgent (Wu et al., 2025b) focus on chemistry, and AgentSwift (Li et al., 2025b) employs hierarchical search without long-horizon simulations. None address iterative temporal dynamics optimization. Language Agent Tree Search (LATS) (Zhou et al., 2023a) and SWE-Search (Antoniades et al., 2024) are closest: LATS operates on discrete task graphs, while SWE-Search targets software engineering. GIF-MCTS (Dainese et al., 2024) and Zhang et al. (2025a) address short sequences. Our method optimizes multi-thousand-step temporal dynamics with tailored representation and search.

## 3 METHOD

### 3.1 PRELIMINARIES

We first formalize the complex systems under discussion and propose a modern programming language-based approach for their representation. Complex systems are typically studied using mathematical models expressed as ordinary differential equations (ODEs) with initial values:

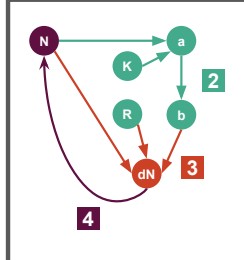 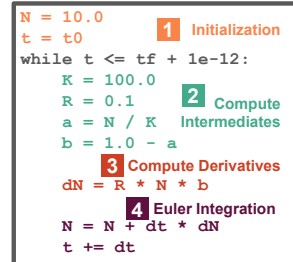 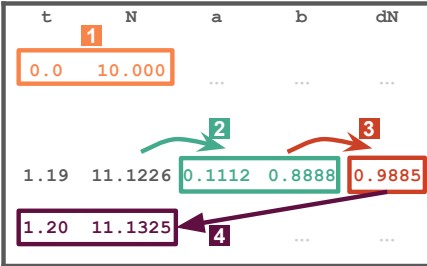

Figure 2: Python representation of a complex system. From left to right: graph representation, Python code, and computation process. Four essential components are shown: (1) initialization of values, (2) intermediate computations, (3) derivative evaluations, and (4) Euler method integration. This representation allows users to easily code and inspect, while enabling LLM understanding.

$$d\boldsymbol{x}/dt = f(\boldsymbol{x}, t), \quad \boldsymbol{x}(t_0) = \boldsymbol{x}_0 \tag{1}$$

where $\boldsymbol{x} \in \mathbb{R}^n$ is the state vector (also called level or stock variables in the complex systems literature), $f : \mathbb{R}^n \times \mathbb{R} \to \mathbb{R}^n$ the vector field, and $\boldsymbol{x}_0 = [x_{0,1}, \quad x_{0,2}, \quad \ldots, \quad x_{0,n}]^\top$ the initial values.

While analytical techniques for ODEs, such as the Laplace transform, can be applied to complex systems analysis when feasible, these systems are more commonly analyzed through numerical approaches like the first-order Euler method (Ogata, 2004) due to the complexity of $f$. This approach enables flexible modeling by requiring only that $f$ be computable, which is crucial because as many complex systems are too intricate for analytical solutions. Concretely, the first-order Euler method discretizes the system with time step $\Delta t$ as:

$$\boldsymbol{x}_{t+\Delta t} = \boldsymbol{x}_t + \Delta t \cdot f(\boldsymbol{x}_t, t), \quad \boldsymbol{x}_0 = \boldsymbol{x}(0) \tag{2}$$

Note that $\boldsymbol{x}$ and its discretized counterparts $\boldsymbol{x}_t$ and $\boldsymbol{x}_{t+\Delta t}$ represent interpretable, meaningful quantities such as populations and resources, rather than abstract hidden states as in RNNs, despite the similarity in their formulations. Historically, communities studying complex systems have benefited from specialized modeling languages like DYNAMO and STELLA (Radzicki & Taylor, 2011; Richmond, 1985). However, as discussed in Section 2, these tools suffer from outdated syntax or proprietary visual interfaces (Forrester, 2011) that limit their modern adoption with LLMs.

### 3.2 UNIFIED REPRESENTATION

To address these limitations, we propose a unified representation based on a restricted subset of Python programs. This choice leverages Python's widespread familiarity and LLMs' strong Python capabilities while avoiding the need to create a new domain-specific language. Practically, this approach enables flexible implementation of $f(\boldsymbol{x}, t)$ through general-purpose programming constructs.

We show an example of the Python representation of an actual complex system in Figure 2. Specifically, we present the system in its graph representation (for reference), the corresponding Python code, and the computation process. The components and computation are divided into four parts: (1) initialization of values ($\boldsymbol{x}_0 = \boldsymbol{x}(0)$), followed by a loop of (2) intermediate computations, (3) derivative evaluations (i.e., computing $f(\boldsymbol{x}_t, t)$), and (4) Euler-method integration (i.e., computing $\boldsymbol{x}_{t+\Delta t}$ from $\boldsymbol{x}_t$ and $f(\boldsymbol{x}_t, t)$). Our implementation features extensive inline documentation and domain-specific mathematical primitives, including wrapper functions for common operations and STELLA-compatible delay and smoothing functions with time-based semantics.

This abstraction layer allows LLMs to manipulate system dynamics through semantic constructs rather than low-level implementation details, prioritizing interpretability while maintaining compatibility with established modeling conventions. The design facilitates both human comprehension and automated LLM-based system generation through standardized interfaces. We provide a complete example showing the syntax and semantics in Appendix A.

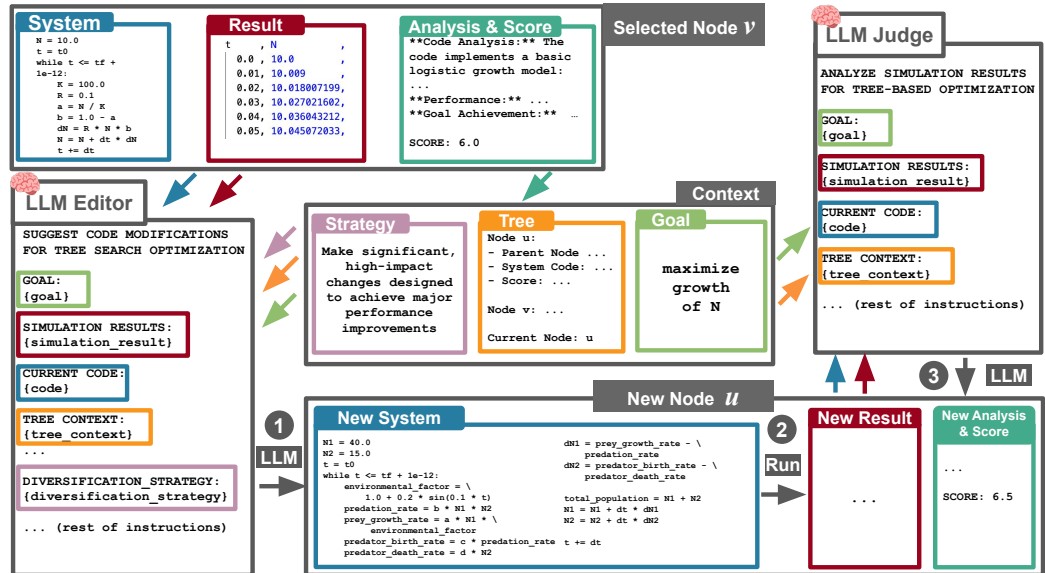

Figure 3: The node expansion process using LLM Editor and Judge. Given a selected node $v$ and expansion strategy $s$, the LLM Editor generates a new system variant $P_u$ by leveraging the parent system $P_v$, its analysis $A_v$, strategy $s$, and goal $G$. The generated system is executed to obtain record $R_u$, then evaluated by the LLM Judge to produce analysis-score pair $(A_u, S_u)$. We embed analysis and scores of all nodes, including the currently selected node, in the tree context to provide the LLM with a comprehensive view of the search tree.

### 3.3 PROPOSED METHOD

We propose CEDAR, an autonomous method that combines Monte Carlo Tree Search (MCTS), large language models, and a unified system representation to discover complex systems that meet specified goals. At each iteration, MCTS selects a system variant for expansion. An LLM Editor then generates improved system variants based on execution feedback. The newly generated variant is subsequently passed to an LLM Judge, which evaluates its performance and provides a comprehensive analysis. We present the overall architecture of CEDAR in Figure 1.

**Formalization** We formalize the key components of CEDAR, illustrated in Figure 1, as follows. We denote a system as $P$, represented using the restricted subset of Python code described above. The execution record $R$ of a given complex system $P$, including the values of all variables at each timestamp, is obtained by $R \leftarrow \text{RUN}(P)$. We define a goal $G$ as a natural-language description of the desired system behavior [1].

The search process employs an MCTS tree $T = (V, E)$ with vertex set $V$ and edge set $E$. For each node $v \in V$, we associate four key components: (1) a system $P_v$, (2) an execution record $R_v \leftarrow \text{RUN}(P_v)$, (3) a bounded numerical score $S_v$, and (4) a textual analysis $A_v$. We denote the parent of node $v$ as parent($v$) and the expansion strategy used to generate $v$ from its parent as $s_v$. The root node is denoted as $0 \in V$ and has neither a parent nor an associated expansion strategy. At each iteration, the LLM Judge produces an analysis-score pair $(A_v, S_v)$ that evaluates the quality of system $P_v$ against the specified goal $G$. Details of the LLM Judge and expansion strategies are discussed in the following sections.

**Initialization** The search process begins by initializing the MCTS tree $T$ with a single root node 0 (where $v_0 = 0$). This root node is associated with an initial complex system $P_0$, which can either be provided by the user or generated as a basic template system. We obtain the execution record $R_0 \leftarrow \text{RUN}(P_0)$ and evaluate the initial system via $(A_0, S_0) \leftarrow \text{LLMJUDGE}(P_0, R_0, G)$, where $S_0 \in \mathbb{R}$ is a bounded numerical score reflecting how well $P_0$ satisfies goal $G$, and $A_0$ is a textual analysis providing qualitative feedback about the system's behavior and potential improvements.

---

[1]For example, goal $G$ can be "`Make the population grow stably`" or include additional information like "`Make the system behavior similar to @load(record.csv)`".

**Node Selection** The node selection process jointly selects a node $v \in V$ for expansion and an expansion strategy $s \in \mathcal{S}$, where $\mathcal{S} = \{s_1, s_2, \ldots, s_k\}$ is a fixed set of expansion strategies. Node selection is based on a selection score $\text{SCORE}(v)$ that combines the node's performance score $S_v$, its depth $\text{DEPTH}(v)$ in the tree, and its expansion count $c_v \in \mathbb{N}$ and its parent $p$'s expansion count: CEDAR uses the generalized UCT score $\text{SCORE}(v) = S_v + \phi(c_v, c_p, \tau, )$. Here, we have $\phi = -\infty$ if $c_v \geq \tau$ and $\alpha \cdot \sqrt{\ln(c_p + 1)/(c_v + 1)} + \gamma \cdot \text{DEPTH}(v)$ otherwise, where $\alpha$ adjust the amount of exploration, $\gamma > 0$ controls the preference for deeper nodes and $\tau$ controls the expansion count. Unlike vanilla MCTS, we limit expandable nodes to a subset $V_{\text{exp}} \subseteq V$ and allow repeated expansion of internal nodes until their expansion count $c_v$ reaches a threshold $\tau \in \mathbb{N}$, inspired by Inoue et al. (2025). Formally, a node $v$ is expandable if and only if $v \in V_{\text{exp}}$ and $c_v < \tau$. The selection process chooses $(v, s) = \arg\max_{v \in V_{\text{exp}}} \text{SCORE}(v)$, with the expansion strategy $s \sim \text{Uniform}(\mathcal{S})$ sampled uniformly. Practically, hyperparameters $\alpha$ and $\tau$ reflect empirical behavior of LLM-based optimization on complex systems: high-quality solutions usually require several consecutive refinements along promising branches, so we favor deeper nodes while capping expansion counts. This cap prevents over-exploring a single subtree and avoids wasting expensive LLM calls.

**Node Expansion with LLMs** Given the selected node $v$ and expansion strategy $s$, the LLM Editor generates a new system variant via $P_u \leftarrow \text{LLMEDITOR}(P_v, A_v, s, G)$. This process leverages the parent system $P_v$, its associated analysis $A_v$, the chosen strategy $s$, and the goal specification $G$ to produce a revised system that satisfies both syntactic constraints and semantic requirements.

The newly generated system $P_u$ is executed to obtain its record $R_u \leftarrow \text{RUN}(P_u)$, followed by performance evaluation $(A_u, S_u) \leftarrow \text{LLMJUDGE}(P_u, R_u, G)$, yielding both a bounded numerical score $S_u \in \mathbb{R}$ and a qualitative analysis $A_u$. Both LLMEDITOR and LLMJUDGE are made feasible through carefully injected semantics and instructions, through two complementary mechanisms: first, specialized prompts that guide the LLM's editing and reasoning process, and second, semantic annotations embedded within the code that help the LLM understand the system's structure and behavior. The full expansion process is illustrated in Figure 3, and details, including extensions in code and prompts, are provided in Appendix B.

A new node $u$ is then added to the tree with the associated system $P_u$, execution record $R_u$, score $S_u$, and analysis $A_u$. This iterative process of selection, expansion, execution, and evaluation enables progressive improvement toward the target objectives through systematic exploration of the system space. We formalize the complete search procedure in Algorithm 1, which provides a detailed algorithmic description of our MCTS-based approach. We also cover further details of our method, including design choices and as well scalability and stability in Appendix C.

---

**Algorithm 1: Monte Carlo Tree Search for Complex System Discovery**

1: **Input:** Initial system $P_0$, goal $G$, iterations $N$, expansion threshold $\tau$
2: **Output:** Best discovered system $P^*$
3: Initialize tree $T = (V, E)$ with root node 0
4: $(A_0, S_0) \leftarrow \text{LLMJUDGE}(P_0, \text{RUN}(P_0), G)$
5: $P^* \leftarrow P_0, S^* \leftarrow S_0$
6: **for** $i = 1$ to $N$ **do**
7:     $(v, s) \leftarrow \arg\max_{v \in V_{\text{exp}}} \text{SCORE}(v)$ where $s \sim \text{Uniform}(\mathcal{S})$
8:     $P_u \leftarrow \text{LLMEDITOR}(P_v, A_v, s, G)$
9:                     $\triangleright$ Generate new system
10:     $R_u \leftarrow \text{RUN}(P_u)$       $\triangleright$ Execute system
11:     $(A_u, S_u) \leftarrow \text{LLMJUDGE}(P_u, R_u, G)$
12:                 $\triangleright$ Evaluate system
13:     Add node $u$ to the tree with $(P_u, R_u, A_u, S_u)$
14:     $c_v \leftarrow c_v + 1$    $\triangleright$ Increment expansion count
15:     **if** $c_v \geq \tau$ **then**
16:         $V_{\text{exp}} \leftarrow V_{\text{exp}} \setminus \{v\}$
17:             $\triangleright$ Remove it from the expandable set
18:     **end if**
19:     **if** $S_u > S^*$ **then**
20:         $P^* \leftarrow P_u, S^* \leftarrow S_u$
21:     **end if**
22: **end for**
23: **return** $P^*$

---

### 3.4 THEORETICAL CONNECTION

While CEDAR components are motivated by practical considerations, the algorithm remains a principled MCTS variant with widening- and depth-aware UCT selection, plus LLM-parameterized transition kernel and value function. While the connections can be established below, we also note that convergence and regret guarantees are open problem for MCTS with LLMs.

**Node Selection as a UCT-Generalization.** Following UCB1 (Kocsis & Szepesvári, 2006), CEDAR uses $\text{SCORE}(v) = S_v + \phi(c_v, c_p, \tau)$, where $c_v$ and $c_p$ are expansion counts at $v$ and its parent $p$, and $\tau$ is the progressive widening limit. We combine two MCTS principles: (1) Progressive Widening (Coulom, 2007; Chaslot et al., 2007; Inoue et al., 2025) limiting expansion width, and (2) depth-based bonuses favoring deeper search (Blackshaw et al., 2025; Wu et al., 2025a). This yields:

$$\phi = \underbrace{\alpha\sqrt{\frac{\ln(c_p + 1)}{c_u + 1}} \cdot \mathbb{I}[c_v < \tau] + \beta\, \mathbb{I}[c_v \geq \tau]}_{\text{Progressive Widening}} + \underbrace{\gamma \cdot \text{DEPTH}(v)}_{\text{Depth-based bonus}}. \tag{3}$$

The node selection corresponds to $\beta = -\infty$, positioning CEDAR a simple combination of MCTS variants with non-uniform branching The constraint $c_v < \tau$ prevents overexpansion in vast action spaces like program edits and expensive LLM invoking.

**LLM Editor as a Learned Transition Kernel.** MCTS assumes a generative model for transitions: $s' \sim P(\cdot \mid s, a)$. CEDAR extends this to a program-valued state space via a stochastic operator $P_\theta$ induced by the LLM:

$$P_u \sim P_\theta(\cdot \mid P_v, A_v, s, G), \quad \text{where } P_\theta = \text{LLMEDITOR}_\theta, \quad s \sim \text{Uniform}(\mathcal{S}) \tag{4}$$

where $P_\theta$ parameterizes a conditional distribution over successor programs. Though parameterized by a pretrained foundation model rather than learned during search, $P_\theta$ functionally serves as a learned component enabling MCTS to operate in complex program spaces. This aligns with recent work using LLMs as learned components: as sampling models (Zhou et al., 2023b; Li et al., 2023), refining models in code synthesis (Chen et al., 2023; Zelikman et al., 2022) and reasoning (Shinn et al., 2023). CEDAR extends LLM-powered transition kernels from linear to tree search, connecting to learned-transition MCTS and model-based RL (Silver et al., 2017b; Schrittwieser et al., 2019), neural-guided search (Chen et al., 2020; Xu et al., 2024), and scientific discovery (Lai & Pu, 2025).

**LLM Judge as a Learned Value Function.** Executing $P_u$ yields a deterministic trajectory $R_u = \text{RUN}(P_u)$. The LLM Judge provides a semantic evaluation:

$$(A_u, S_u) = R_\phi(P_u, R_u, G), \quad \text{where } R_\phi = \text{LLMJUDGE}_\phi \tag{5}$$

where $R_\phi$ acts as a learned value network. Focusing on scoring, we denote $S_u = \pi_S(R_\phi(P_u, R_u, G))$ where $\pi_S$ projects from $(A, S)$ onto score $S$. Recent work shows LLMs can serve as noisy reward estimators or constraint evaluators in non-differentiable optimization (Zhou et al., 2023b; Zelikman et al., 2022; Shinn et al., 2023), often pairing with LLM also serving as sampling models. Theoretically, $S_u$ constitutes a *bounded, noisy reward*, falling under convergence analysis of MCTS with biased evaluators (Lisy et al., 2013; Efroni et al., 2019). UCT remains consistent when reward noise is bounded and non-adversarial, as with semantic LLM scoring.

**Theoretical Guarantees Remains an Open Problem** Unlike classical MCTS with well-specified MDPs where asymptotic convergence and regret guarantees are established Kocsis & Szepesvári (2006); Bubeck et al. (2011), Recent work combining MCTS with LLM yields strong emperical performances on program related tasks Li et al. (2024; 2025a); Xu et al. (2025) while the focus is not on theoretical analysis Katz et al. (2024) The current theoretical work only provides at best partial guarantees (e.g., lose verifier-induced upper bounds Brandfonbrener et al. (2024)), leaving the general problems of *optimality* and *convergence* for MCTS and LLM methods open research questions.

## 4 EXPERIMENTS

CEDAR enables three previously infeasible capabilities: optimizing for vague natural-language goals, fitting records without complete system skeletons, and providing interpretable optimization. We present qualitative demonstrations and quantitative comparisons.

### 4.1 EXPERIMENTAL DETAILS

**Data.** We collect and convert models from classical works in complex systems in our modern framing. We convert *World Dynamics* (Forrester, 1971) from the DYNAMO language and 19 systems

from the book Modeling Dynamic Biological Systems (Hannon & Ruth, 2014) from the STELLA language. We utilize these systems in two ways: as foundations to be further optimized for multiple goals, and as ground truth systems that produce reference records serving as optimization targets.

**Experimental Setup.** We use CEDAR with implementation details provided in the appendix. Our MCTS configuration allows 20 node selections, with each selection expanding to 5 new candidate nodes, resulting in a total of 100 expansions per search. We mainly use LLM providers: Anthropic Claude Sonnet 4.5 as LLM provider for its powerful coding capacity, and emperically set hyperparameters $\alpha = 1$ and $\gamma = 2$ for node selection. The expansion strategy is selected by sampling from a set of candidate strategies. Details regarding the statistics of complex systems, expansion strategies and computational costs are covered in Appendix D.

### 4.2 FITTING AN ABSTRACT GOAL DESCRIBED IN NATURAL LANGUAGE

In this experiment, we present a comprehensive walkthrough of the optimization process with concrete examples. The system we utilize is *World Dynamics*, which is the complex system with the most variables in our collection. This complex system models the co-evolution of human population, resource utilization, and pollution as a nonlinear, feedback-driven system. Our ambitious goal is to jointly balance population growth, resource usage, and pollution. Figure 4 illustrates the MCTS search tree and the resulting system dynamics. The baseline, *resulting, published system* from the monograph Forrester (1971), is a complex system designed by hand, good enough such that further optimizing for one sub-goal often comes at the expense of others. We show this in Appendix E to highlight the intrinsic challenges of optimizing complex systems with competing objectives. , this visualization demonstrates how the search tree effectively further guides the optimization process toward solutions with higher scores. The final evolved system satisfies the vague goal, showing meaningful improvements over the initial configuration, with all three key variables demonstrating enhanced performance. Interestingly, with different backends, CEDAR discovers multiple structurally distinct but high-performing systems, each emphasizing different tradeoffs among objectives (for example, prioritizing population by Claude versus resource preservation by GPT-5.1). The full natural-language goal specification is detailed in Appendix E.

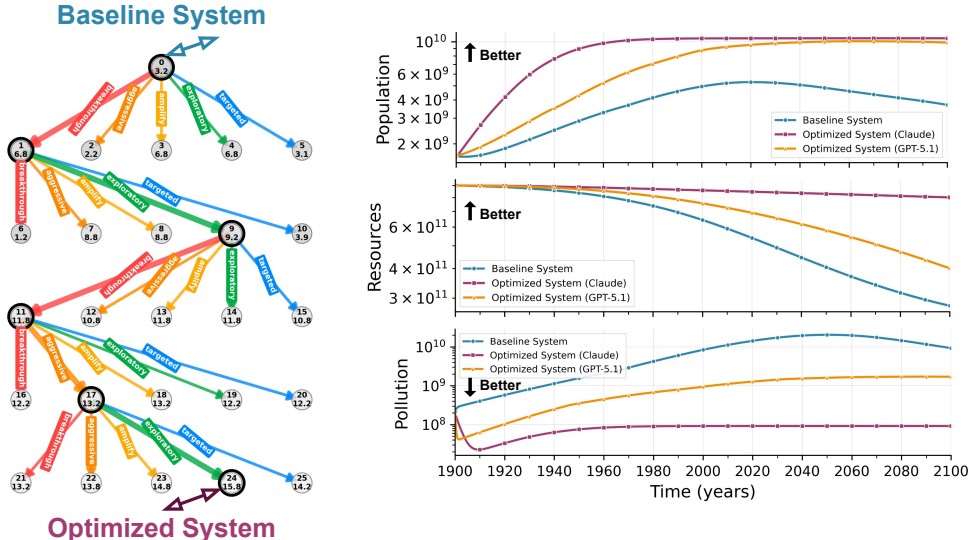

Figure 4: Illustration of fitting an abstract, language-based goal from *World Dynamics*. On the left, the MCTS tree is shown with the best expansion path highlighted. Each node is marked by the ID and LLM Judge Score. On the right, we display the system dynamics for both the initial (original) and final optimized systems, including Population ($\uparrow$), Pollution ($\downarrow$), and Resources ($\uparrow$). With CEDAR, the system reaches better states (increased population, fewer resources used, and less pollution).

### 4.3 Quantitative Studies - Fitting a Concrete Record

In this experiment, we task CEDAR with evolving a system (using only a bare-minimum skeleton) to fit the simulated record of a more complex model. While the strength of CEDAR lies in fitting vague, abstract goals, it can nonetheless be used to fit records from either observations or ground truth systems. To do so, we simply load the record into the goal and run CEDAR as-is without further modification, leveraging its universal design. We compare CEDAR with existing approaches based on black-box optimization. The ground truth is a population growth model with stochastic components. We use Optuna as our black box optimization baseline and employ both L1 and Dynamic Time Warping (DTW) distances as the scoring metric. Since the search space for black box optimization consists of only coefficients, we explore different formulae to support the baseline: no formulae, a simple version, or the full formulae in the ground truth system, the last of which gives the baseline a serious boost. Even so, as shown in Table 1 and Figure 5, we find that Optuna's performance is severely limited by the skeletal formulae, while CEDAR, even without such predefined formulations, is capable of finding solutions better than Optuna, which has help of full formulae skeleton. The no/simple-formula baselines remain clearly worse both quantitatively and qualitatively. The details regarding the DTW, ground truth system, supporting formulae, and Optuna on simple baseline are provided in the Appendix F.

Table 1: L1 Distance and DTW comparison. CEDAR achieves better performance without formulae than black-box optimization with full formulae.

| Method | Formulae | | | L1 | DTW |
| --- | --- | --- | --- | --- | --- |
| | No | S. | F. | | |
| Optuna | ✓ | | | 29.06 | 5503.68 |
| Optuna | | ✓ | | 26.01 | 4927.94 |
| Optuna | | | ✓ | 3.71 | 477.52 |
| CEDAR (Claude) | ✓ | | | 3.29 | 757.21 |
| CEDAR (GPT-5.1) | ✓ | | | **2.22** | **433.13** |

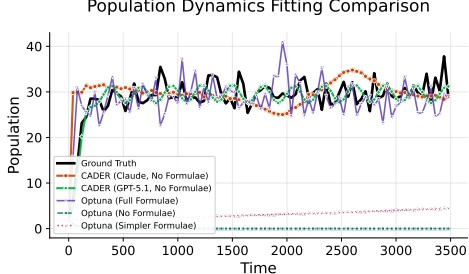

Figure 5: Comparison of system dynamics between CEDAR and the Optuna baseline when fitting concrete records.

### 4.4 Interpretability

For the *World Dynamics* system in Section 4.2, we demonstrate interpretability by showing how the LLM's analysis of candidate systems interacts with our search method. We highlight, in Table 2, the LLM responses along the path that produces the best system in the search tree shown in Figure 4. CEDAR provides clear, human-understandable interpretability of three aspects in the goal to be balanced. The LLM Judge analyzes issues with a balanced focus on the aspects specified in the goal, while the LLM Editor first evaluates general system performance and then fine-tunes the individual aspects, also attending to components that are not explicitly mentioned (e.g., capital productivity). This interpretability significantly reduces the human effort required to understand the optimization.

### 4.5 Why MCTS: Diversity of Solutions and Better Performance

We find that the benefits of MCTS are three-fold. **First, MCTS optimizes complex systems while preserving solution diversity.** We assess diversity by focusing on the complex system discussed in Section 4.3. Within a single search tree, we sample nodes whose scores are close to the best solution and visualize each node's population trajectory after the initial rapid growth phase (i.e. $t > 100$) in Figure 6. Although all systems satisfy the goal, they follow distinct trajectories. Unlike the Optuna-driven baselines in Section 4.3, CEDAR avoids collapsing solutions to a single point that resembles overfitting. This diversity enables better decision-making through sensitivity analysis.

**The second benefit is better performance.** This appears most clearly in ablation studies that evaluate the contribution of MCTS to CEDAR. We compare MCTS against linear search to demonstrate the importance of tree-based exploration and illustrate the differences between the two strategies (node score distributions and resulting population dynamics) in Figure 7. MCTS yields higher-scoring

Table 2: LLM Editor and Judge responses along the trajectory to the best solution. Population, pollution, and resources are manually colored for visibility. In this table, we show the LLM summarized version. The relation between LLM analysis and LLM scores and the full transcripts of responses, which include detailed analysis and editing rationales, are provided in Appendix G.

| Depth | LLM Editor Response | LLM Judge Response |
|---|---|---|
| 0 (root) | (No editor response for root, which is the initial system given as is.) | Unsustainable overshoot behavior – 69% resource depletion, 46x pollution increase |
| 1 | Breakthrough approach – 25-40% efficiency improvements targeting 6-8 score | Sustainable progress – population to 5.46B, 30% resources remaining, controlled pollution |
| 2 | Exploratory approach – radical efficiency through enhanced capital productivity | Paradigm shift – 7.0B population, 78% resource conservation, near-zero pollution |
| 3 | Ultra-aggressive conservation – pushing toward exceptional 9.5+ performance | Breakthrough sustainability – 3.2x population, 36% resource depletion, 90% pollution improvement |
| 4 | Revolutionary breakthrough – 75% resource reduction, 90% pollution control, 300% capital productivity | Best-in-tree performance – 6.3x population, 59% resource conservation, exceptional pollution control |
| 5 | Extreme efficiency breakthrough – 95% resource conservation, 98% pollution reduction, 15x capital productivity | Holy grail achievement – 6.4x population growth with only 19.9% resource depletion |

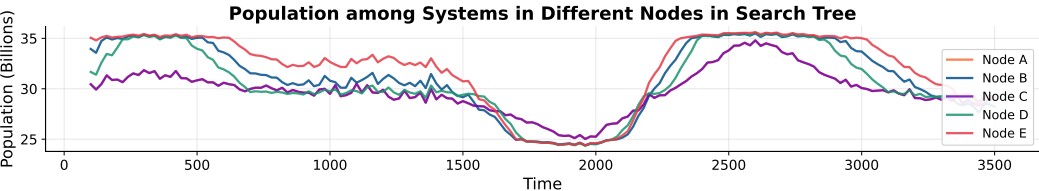

Figure 6: Diversity of solutions showing different population trajectories after the system passes the initial phase in the search tree for the complex system in Section 4.3.

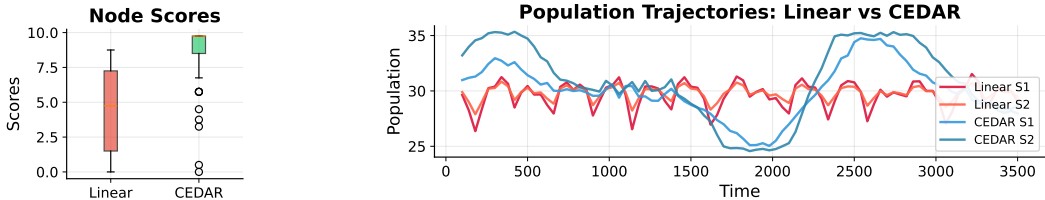

Figure 7: Comparison between MCTS and linear search showing performance differences in search strategy and resulting population dynamics.

nodes and smoother trajectories that avoid overfitting-like behavior. This shows that our design choice to combine MCTS with LLMs is crucial for performance gains.

**The third benefit is LLM provider comparability.** Our design separates optimization logic from tree search structure, enabling users to specify different LLM providers while maintaining the same framework. This supports comparing providers under identical conditions. Structurally distinct high-performing systems emerge in Section 4.2 and 4.3, each emphasizing different tradeoffs. This facilitates systematic evaluation of how foundation models affect optimization outcomes.

## 5 CONCLUSION

We propose CEDAR, combining Monte Carlo Tree Search with large language models to optimize complex systems for vague, natural-language goals, outperforming black-box optimization without predefined formulae. The unified Python representation provides interpretable insights and diverse solutions. Experiments demonstrate CEDAR's capabilities across diverse complex-systems domains, opening new possibilities where traditional approaches struggle. We discuss future directions on technical improvements and new human-in-the-loop paradigm in Appendix H.

## ETHICS STATEMENT

CEDAR streamlines optimization of complex dynamic systems through automated reasoning and iterative refinement. It applies across domains such as resource allocation, scheduling, and strategic planning. By supporting flexible objectives and constraints, it addresses problems that were previously difficult to solve, improving efficiency and effectiveness.

This study involves no human subjects or sensitive data and focuses solely on optimization methodology. Downstream use should be thoughtful and oriented toward positive impact, with reasonable safeguards and oversight to assess potential effects on individuals and society in real-world deployments.

## REPRODUCIBILITY STATEMENT

To ensure reproducibility, we include the full MCTS algorithm in the main text and detailed prompts for the LLM Editor and Judge components in the appendix. Because LLM behavior can change as providers update models, we provide complete transcripts of LLM interactions throughout the optimization process for verification. We also provide the complete implementation code and the complex systems used in our experiments in the supplementary materials.

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

# APPENDIX

## A   PYTHON IMPLEMENTATION OF COMPLEX SYSTEMS

This section presents the Python implementation of our unified representation of complex systems, which comprises four essential components: (1) initialization, (2) intermediate computations, (3) derivative evaluations, and (4) Euler integration. A minimal example illustrating the core structure is shown below:

```python
# (1) initialization
dt = 0.01
t0 = 0.0
tf = 120.0
K = 100.0
R = 0.1
N = 10.0

t = t0
while t <= tf + 1e-12:

    # (2) intermediate computations
    carrying_capacity_factor = 1.0 - N / K
    growth_rate = R * N * carrying_capacity_factor

    # (3) derivative evaluations
    dN = growth_rate   # dN = R * N * (1 - N/K)

    # (4) Euler integration
    N = N + dt * dN
    t += dt
```

In practice, we enrich the code with comprehensive comments for both human interpretation and LLM comprehension. This enables LLMs to understand system semantics, guiding optimization while enforcing key constraints. Several critical restrictions must be respected, such as preserving the Euler integration scheme, for optimization to execute successfully. Below, we present the *World Dynamics* system (Forrester, 1971) (one with the largest by number of variables in our dataset), represented using the restricted Python framework.

```python
# ==================== BEGIN CONFIG ====================
# Simulation configuration parameters (DO NOT CHANGE - these are set by the system)
dt = 0.2   # time step size
t0 = 1900.0   # start time
tf = 2100.0   # end time
seed = 1234   # random seed (set None for nondeterministic runs)
# ==================== END CONFIG ====================

# ==================== BEGIN STATE ====================
# STATE VARIABLES - integrated over time (must define derivatives below)
P = 1.65e9   # population (people) # NECESSARY
NR = 900e9   # natural resources (natural resource units) # NECESSARY
CI = 0.4e9   # capital investment (capital units) # NECESSARY
POL = 0.2e9   # pollution (pollution units) # NECESSARY
CIAF = 0.2   # capital-investment-in-agriculture fraction (dimensionless) # NECESSARY
# ==================== END STATE ====================

# ================ BEGIN ALGEBRAIC (OPTIONAL) ================
# Variables are now declared in the HELPERS section within the simulation loop

# ==================== BEGIN PARAMS ====================
# CONSTANT PARAMETERS used in equations
# (All constants are inlined in the code)
# ==================== END PARAMS ====================

# ==================== MAIN LOOP ====================
t = t0
while t <= tf + 1e-12:
    # ==================== BEGIN INPUTS_T ====================
    # TIME-DEPENDENT INPUTS - use wrappers like sin(), exp(), graph()
    # (No time-dependent inputs in this model)
    # ==================== END INPUTS_T ====================

    # ================ BEGIN INPUTS_RND ====================
    # RANDOM INPUTS - use gauss(), uniform() wrappers (no STATE usage).
    # (No random inputs in this model)
```

```python
37      # =================== END INPUTS_RND =======================
38
39      # =================== BEGIN HELPERS =======================
40      # HELPERS - All computed variables (algebraic and intermediate expressions)
41      # in dependency order.
42
43      # Basic ratios and derived quantities
44      CIR = CI / P  # capital-investment ratio (capital units/person)
45      CR = P / (135e6 * 26.5)  # crowding ratio (dimensionless)
46      NRFR = NR / 900e9  # natural-resource fraction remaining (dimensionless)
47      POLR = POL / 3.6e9  # pollution ratio (dimensionless)
48
49      # Material standard of living components
50      CIAF_current = CIAF  # current capital-investment-in-agriculture fraction
51      NREM = graph(
52          NRFR, ((0, 0), (0.25, 0.15), (0.5, 0.5), (0.75, 0.85), (1, 1))
53      )  # natural-resource-extraction multiplier
54      ECIR = CIR * (1 - CIAF_current) * NREM / (1 - 0.3)  # effective-capital-investment
              ratio
55      MSL = ECIR / 1  # material standard of living (dimensionless)
56
57      # Birth rate multipliers
58      BRMM = graph(
59          MSL, ((0, 1.2), (1, 1), (2, 0.85), (3, 0.75), (4, 0.7), (5, 0.7))
60      )  # birth-rate-from-material multiplier
61      BRCM = graph(
62          CR, ((0, 1.05), (1, 1), (2, 0.9), (3, 0.7), (4, 0.6), (5, 0.55))
63      )  # birth-rate-from-crowding multiplier
64      BRPM = graph(
65          POLR, ((0, 1.02), (10, 0.9), (20, 0.7), (30, 0.4), (40, 0.25), (50, 0.15), (60,
                  0.1))
66      )  # birth-rate-from-pollution multiplier
67
68      # Food ratio components
69      CIRA = CIR * CIAF_current / 0.3  # capital-investment ratio in agriculture
70      FPCI = graph(
71          CIRA, ((0, 0.5), (1, 1), (2, 1.4), (3, 1.7), (4, 1.9), (5, 2.05), (6, 2.2))
72      )  # food potential from capital investment
73      FCM = graph(CR, ((0, 2.4), (1, 1), (2, 0.6), (3, 0.4), (4, 0.3), (5, 0.2)))  # food-
              from-crowding multiplier
74      FPM = graph(
75          POLR, ((0, 1.02), (10, 0.9), (20, 0.65), (30, 0.35), (40, 0.2), (50, 0.1), (60,
                  0.05))
76      )  # food-from-pollution multiplier
77      FR = FPCI * FCM * FPM * (1 if t >= 1970 else 1) / 1  # food ratio
78      BRFM = graph(FR, ((0, 0), (1, 1), (2, 1.6), (3, 1.9), (4, 2)))  # birth-rate-from-food
              multiplier
79
80      # Death rate multipliers
81      DRMM = graph(
82          MSL,
83          (
84              (0, 3),
85              (0.5, 1.8),
86              (1, 1),
87              (1.5, 0.8),
88              (2, 0.7),
89              (2.5, 0.6),
90              (3, 0.53),
91              (3.5, 0.5),
92              (4, 0.5),
93              (4.5, 0.5),
94              (5, 0.5),
95          ),
96      )  # death-rate-from-material multiplier
97      DRCM = graph(CR, ((0, 0.9), (1, 1), (2, 1.2), (3, 1.5), (4, 1.9), (5, 3)))  # death-
              rate-from-crowding multiplier
98      DRPM = graph(
99          POLR, ((0, 0.92), (10, 1.3), (20, 2), (30, 3.2), (40, 4.8), (50, 6.8), (60, 9.2))
100     )  # death-rate-from-pollution multiplier
101     DRFM = graph(
102         FR, ((0, 30), (0.25, 3), (0.5, 2), (0.75, 1.4), (1, 1), (1.25, 0.7), (1.5, 0.6),
                  (1.75, 0.5), (2, 0.5))
103     )  # death-rate-from-food multiplier
104
105     # Natural resource usage multiplier
106     NRMM = graph(
107         MSL, ((0, 0), (1, 1), (2, 1.8), (3, 2.4), (4, 2.9), (5, 3.3), (6, 3.6), (7, 3.8),
                  (8, 3.9), (9, 3.95), (10, 4))
108     )  # natural-resource-from-material multiplier
109
```

```
110      # Capital investment multiplier
111      CIM = graph(MSL, ((0, 0.1), (1, 1.0), (2, 1.8), (3, 2.4), (4, 2.8), (5, 3)))  # capital
             -investment multiplier
112
113      # Pollution components
114      POLCM = graph(CIR, ((0, 0.05), (1, 1), (2, 3), (3, 5.4), (4, 7.4), (5, 8)))  #
             pollution-from-capital multiplier
115      POLAT = graph(
116          POLR, ((0, 0.6), (10, 2.5), (20, 5), (30, 8), (40, 11.5), (50, 15.5), (60, 20))
117      )  # pollution-absorption time
118
119      # Quality of life components
120      QLM = graph(MSL, ((0, 0.2), (1, 1), (2, 1.7), (3, 2.3), (4, 2.7), (5, 2.9)))  # quality
             of life from material
121      QLC = graph(
122          CR,
123          (
124              (0, 2),
125              (0.5, 1.3),
126              (1, 1),
127              (1.5, 0.75),
128              (2, 0.55),
129              (2.5, 0.45),
130              (3, 0.38),
131              (3.5, 0.3),
132              (4, 0.25),
133              (4.5, 0.22),
134              (5, 0.2),
135          ),
136      )  # quality of life from crowding
137      QLF = graph(FR, ((0, 0), (1, 1), (2, 1.8), (3, 2.4), (4, 2.7)))  # quality of life from
             food
138      QLP = graph(
139          POLR, ((0, 1.04), (10, 0.85), (20, 0.6), (30, 0.3), (40, 0.15), (50, 0.05), (60,
             0.02))
140      )  # quality of life from pollution
141      QL = 1 * QLM * QLC * QLF * QLP  # quality of life
142
143      # Capital investment fraction adjustment components
144      CFIFR = graph(FR, ((0, 1), (0.5, 0.6), (1, 0.3), (1.5, 0.15), (2, 0.1)))  # capital
             fraction indicated by food ratio
145      CIQR = graph(QLM / QLF, ((0, 0.7), (0.5, 0.8), (1, 1), (1.5, 1.5), (2, 2)))  # capital-
             investment-from-quality ratio
146
147      # ==================== END HELPERS ========================
148
149      # ================== BEGIN DERIVATIVES ====================
150      # DERIVATIVES for each STATE variable.
151
152      # Birth and death rates
153      BR = P * (0.04 if t >= 1970 else 0.04) * BRFM * BRMM * BRCM * BRPM  # birth rate
154      DR = P * (0.028 if t >= 1970 else 0.028) * DRMM * DRPM * DRFM * DRCM  # death rate
155
156      # Natural resource usage rate
157      NRUR = P * (1 if t >= 1970 else 1) * NRMM  # natural-resource-usage rate
158
159      # Capital investment flows
160      CIG = P * CIM * (0.05 if t >= 1970 else 0.05)  # capital-investment generation
161      CID = CI * (0.025 if t >= 1970 else 0.025)  # capital-investment discard
162
163      # Pollution flows
164      POLG = P * (1 if t >= 1970 else 1) * POLCM  # pollution generation
165      POLA = POL / POLAT  # pollution absorption
166
167      # State derivatives
168      dP = BR - DR
169      dNR = -NRUR
170      dCI = CIG - CID
171      dPOL = POLG - POLA
172      dCIAF = (1 / 15) * (CFIFR * CIQR - CIAF)
173      # ================== END DERIVATIVES ====================
174
175      # --------------- Euler integration (engine; do not edit) ----------------
176      P = P + dt * dP
177      NR = NR + dt * dNR
178      CI = CI + dt * dCI
179      POL = POL + dt * dPOL
180      CIAF = CIAF + dt * dCIAF
181
182      # Advance time
183      t += dt
```

972
973
974
975
976
977
978
979

# B  Node Expansion Details

## B.1  Enriching Python Code for Complex Systems

To improve LLM understanding and tool use, we extend the code with extensive inline documentation and predefined mathematical primitives (trigonometric functions, delay operators, smoothing functions, and lookup tables). This design supports both human comprehension and automated LLM-based system generation and modification via clear semantic annotations and standardized function interfaces. Furthermore, this approach ensures the code remains self-contained and executable. The extended code is shown below. Complex systems are embedded between `BEGIN WHOLE SYSTEM` and `END WHOLE SYSTEM`.

```python
# ===============================================================
# sim_base.py - Minimal Euler simulator with edit markers
#
# PURPOSE:
# - Self-contained forward Euler simulator:
#        X(t+dt) = X(t) + dt * dX/dt
#
# HOW TO EDIT:
# - Only change code between the "BEGIN ... / END ..." markers below.
# - Do NOT modify imports, wrappers, loop structure, Euler integration, or OUTPUT section.
# - Keep STATE, ALGEBRAIC, HELPERS, and DERIVATIVES consistent.
# - NEVER change simulation configuration parameters (dt, t0, tf, seed) in the CONFIG
#     section.
#
# VARIABLE MODIFICATION RULES:
# - STATE VARIABLES: You can add new state variables. You can delete state variables ONLY
#     if they
#   are not marked as "# NECESSARY" in comments. All state variables must have derivatives.
# - ALGEBRAIC VARIABLES: You can add new algebraic variables. You can delete algebraic
#     variables
#   ONLY if they are not marked as "# NECESSARY" in comments.
# - HELPERS: You can freely add, remove, and modify helper variables. You can change their
#   computation formulas and reorder them, as long as the code remains valid.
# - When converting from other systems: Mark essential state and algebraic variables with
#   "# NECESSARY" comments so optimization/editing knows which variables cannot be deleted.
#
# VARIABLE TYPES:
# - STATE: variables integrated over time (must have derivatives).
# - INPUTS_T: pure functions of time (cannot depend on STATE).
# - INPUTS_RND: fresh random values each step (cannot depend on STATE).
# - HELPERS: all computed variables (algebraic and intermediate expressions)
#            in dependency order. Can use wrappers like sin(), exp(), graph(),
#            delay(), smth1() - e.g. nonlinear damping from a table.
# - DERIVATIVES: d/dt for each STATE variable.
#
# TRACE CONTENT (per step, in fixed order):
#   1) t               - current time
#   2) STATE variables
#   3) HELPERS (all computed variables)
#   4) DERIVATIVES
#
# OUTPUT:
# - If --csv <path> is provided -> write full simulation to that CSV file.
# - Otherwise -> print the raw CSV (header + all rows) to stdout.
# - Columns/ordering come from the snapshot dict defined inside the loop.
# ===============================================================

import math
import random
import argparse
import pandas as pd  # used only for CSV export
from typing import Iterable, Tuple

# ------------------ Wrapper Functions -----------------------
# These wrappers hide Python semantics; LLMs should ONLY call these.

def sin(x: float) -> float:
    """Sine function (angle in radians)."""
    return math.sin(x)

def cos(x: float) -> float:
```

```python
59        """Cosine function (angle in radians)."""
60        return math.cos(x)
61
62   def exp(x: float) -> float:
63        """Exponential function e^x."""
64        return math.exp(x)
65
66   def tanh(x: float) -> float:
67        """Hyperbolic tangent."""
68        return math.tanh(x)
69
70   def sqrt(x: float) -> float:
71        """Square root."""
72        return math.sqrt(x)
73
74   def log(x: float) -> float:
75        """Natural logarithm."""
76        return math.log(x)
77
78   def gauss(std: float) -> float:
79        """Gaussian random variable with mean=0 and standard deviation=std."""
80        return random.gauss(0.0, std)
81
82   def uniform(lo: float, hi: float) -> float:
83        """Uniform random variable between lo and hi."""
84        return random.uniform(lo, hi)
85
86   def graph(v: float, table: Iterable[Tuple[float, float]]) -> float:
87        """
88        Lookup helper with linear interpolation.
89        - table is an iterable of (x, y) points sorted by x.
90        - If v < x0, return y0.
91        - If v > xN, return yN.
92        - Else, linearly interpolate between nearest points.
93        """
94        table = list(table)
95        if not table:
96            raise ValueError("graph() called with empty table")
97
98        if v <= table[0][0]:
99            return table[0][1]
100       if v >= table[-1][0]:
101           return table[-1][1]
102
103       for i in range(len(table) - 1):
104           x0, y0 = table[i]
105           x1, y1 = table[i + 1]
106           if x0 <= v <= x1:
107               if x1 == x0:
108                   return y0
109               frac = (v - x0) / (x1 - x0)
110               return y0 + frac * (y1 - y0)
111       return table[-1][1]
112
113  # ------------------ Delay/Smooth System ---------------------
114  # STELLA-compatible delay/smooth functions with time-based semantics.
115  # - All functions require a 'name' parameter for unique identification
116  # - Updates occur when functions are called during helper computation
117  # - Fixed delays use time-indexed history with linear interpolation
118  # - Smooth functions use stock-based integration with proper dt scaling
119  # - All functions are dt-independent (same behavior regardless of step size)
120
121  _delay_states = {}  # Registry for all delay/smooth function states
122
123  def delay(input_val: float, delay_time: float, name: str, initial_value: float = None) ->
          float:
124       """Fixed lag delay - returns input value from delay_time ago."""
125       key = f"delay_{name}_{delay_time}"
126
127       if key not in _delay_states:
128           init_val = input_val if initial_value is None else initial_value
129           _delay_states[key] = {
130               'history': [(t, init_val)],
131               'last_t': t
132           }
133
134       state = _delay_states[key]
135
136       # Add current input to history (only if time advanced)
137       if t > state['last_t']:
138           state['history'].append((t, input_val))
```

```python
139            state['last_t'] = t
140
141            # Clean old history beyond delay time
142            cutoff_time = t - delay_time
143            state['history'] = [(t_hist, val) for t_hist, val in state['history']
144                                if t_hist >= cutoff_time]
145
146        # Find value at t - delay_time using linear interpolation
147        target_time = t - delay_time
148        if not state['history'] or target_time <= state['history'][0][0]:
149            return state['history'][0][1]
150
151        for i in range(len(state['history']) - 1):
152            t1, v1 = state['history'][i]
153            t2, v2 = state['history'][i + 1]
154            if t1 <= target_time <= t2:
155                if t2 == t1:
156                    return v1
157                frac = (target_time - t1) / (t2 - t1)
158                return v1 + frac * (v2 - v1)
159
160        return state['history'][-1][1]
161
162 def smth1(input_val: float, smooth_time: float, name: str, initial_value: float = None) ->
         float:
163     """First-order exponential smooth - stock-based smoothing process."""
164     key = f"smth1_{name}_{smooth_time}"
165
166     if key not in _delay_states:
167         init_val = input_val if initial_value is None else initial_value
168         _delay_states[key] = {
169             'smooth_of_input': init_val,  # The stock being smoothed
170             'last_t': t
171         }
172
173     state = _delay_states[key]
174
175     if t > state['last_t']:
176         dt_step = t - state['last_t']
177
178         # SMTH1 equations from STELLA:
179         # Change_in_Smooth = (Input - Smooth_of_Input) / Averaging_Time
180         change_in_smooth = (input_val - state['smooth_of_input']) / smooth_time if
                 smooth_time > 0 else 0.0
181
182         # Stock integration: Smooth_of_Input = Smooth_of_Input + dt * Change_In_Smooth
183         state['smooth_of_input'] += dt_step * change_in_smooth
184         state['last_t'] = t
185
186     # SMTH1 returns the smoothed stock value
187     return state['smooth_of_input']
188
189
190 # ==================== BEGIN WHOLE SYSTEM ====================
191 #
192 #
193 #
194 # ==================== END WHOLE SYSTEM ====================
195
196 # ========================= OUTPUT (FIXED) ====================
197 # DO NOT EDIT THIS SECTION.
198 def _parse_args():
199     ap = argparse.ArgumentParser(description="Run Euler simulation and export CSV.")
200     ap.add_argument("--csv", dest="csv_path", default=None,
201                     help="Write results to this CSV path. If omitted, prints CSV to stdout.
                         ")
202     return ap.parse_args()
203
204 if __name__ == "__main__":
205     args = _parse_args()
206     df = pd.DataFrame(trace)  # column order = insertion order of dict
207     if args.csv_path:
208         df.to_csv(args.csv_path, index=False, float_format='%.6f')
209         print(f"Simulation complete. Results written to {args.csv_path}")
210     else:
211         print(df.to_csv(index=False, float_format='%.6f'), end="")
```

## B.2 Prompt Design and Context Presentation

The effectiveness of CEDAR relies on well-crafted prompts that enable LLMs to understand system dynamics and generate meaningful modifications. We employ specialized prompts for two key components: the LLMEDITOR for system modification and the LLMJUDGE for performance evaluation. These prompts incorporate domain-specific instructions, constraint specifications, and contextual information to guide LLM reasoning effectively.

---

**Prompt for LLMEDITOR**

```
SUGGEST CODE MODIFICATIONS FOR TREE SEARCH OPTIMIZATION

GOAL:
{goal}

CURRENT CODE:
{code}

SIMULATION RESULTS:
{csv_data}

TREE CONTEXT:
{tree_context}

=== For your reference ===

Format for code in tree context:
The tree context shows only the differences in each reference
node compared to current code.
Since the full current code is shown above, diffs only display
lines that are different in reference nodes:
- Lines starting with "+" show what the reference node has
instead of current code
- "Code identical to current node" means no differences exist

How to modify the code to better achieve the goal within the
tree search context:
Focus on:
- Parameters within the marked BEGIN/END edit sections
- State variables, constants, and simulation configuration
- Derivative equations and helper expressions
- Table values and time-dependent inputs
- Learning from tree context and reference node outcomes

When you modify the code:
- Faithfully follow the CURRENT CODE. Starting from EVERYTHING
in CURRENT CODE, then
- only modify code between the "BEGIN ... / END ..." markers.
- DO NOT change imports, wrappers, loop structure, or OUTPUT
section.

Tree Search Strategy:
- Consider your position in the search tree
- Learn from performance patterns in tree context
- Use diversification strategy from tree context
- Reference successful/failed approaches from other nodes

SCORING GUIDELINES:
ABSOLUTE PERFORMANCE SCORING (depth-constrained scale)
```

---

```
    Primary Principle: Score based on how well this code achieves
    the optimization goal, regardless of other nodes in the tree.

    SCORING SCALE (use 2 decimal places like 7.25, 12.75):
    - 0.00-2.00: Failed progress - no meaningful progress toward the
     goal
    - 2.00-4.00: Partial progress - some improvement but far from
    achieving the goal
    - 4.00-6.00: Moderate progress - measurable improvement and
    moving toward the goal
    - 6.00-8.00: Good progress - clear advancement with substantial
    goal achievement
    - 8.00-10.00: Excellent progress - meets most requirements of
    the optimization goal
    - 10.00+: VERY GOOD performance - exceeds the goal expectations
    significantly
    - 20.00+: EXCEPTIONAL performance - far surpasses what the goal
    was asking for

    IMPORTANT CONSTRAINT: Maximum possible score is score <= 10.0 +
    2.5 * depth. Within this limit, high scores are encouraged when
    performance truly merits them.

    HOW TO SCORE:
    1. First, evaluate how well this code achieves the goal in
    absolute terms
    2. Use the tree context only to calibrate what score ranges mean
     - don't let it constrain your scoring
    3. For very good performance that exceeds expectations, don't
    hesitate to give high scores
    4. When unsure between score ranges, favor the higher score if
    genuine progress is evident
    5. Large score increases (5+ points) are appropriate for
    significant improvements
    6. Always respect the depth constraint: score <= 10.0 + 2.5 *
    depth

    Remember: Judge this code's actual achievement of the goal, not
    its relative position in the search tree.

    === Task ====

    Please provide your response in this format:

    REASONING: [Start with a concise description of your main
    modification strategy in the first 200 characters, then explain
    detailed reasoning and assess the expected improvement using the
     scoring guidelines above]

    SELF_ASSESSED_SCORE: [numerical score based on the scoring
    guidelines above for your expected improvement]

    MODIFIED_CODE: [Complete modified Python code following all
    requirements - start from EVERYTHING in CURRENT CODE, then only
    modify code between BEGIN/END markers]
```

### Prompt for LLMJUDGE

```
    ANALYZE SIMULATION RESULTS FOR TREE-BASED OPTIMIZATION
```

```
GOAL:
{goal}

CURRENT CODE:
{code}

SIMULATION RESULTS:
{csv_data}

TREE CONTEXT:
{tree_context}

=== For your reference ===

Format for code in tree context:
The tree context shows only the differences in each reference
node compared to current code.
Since the full current code is shown above, diffs only display
lines that are different in reference nodes:
- Lines starting with "+" show what the reference node has
instead of current code
- "Code identical to current node" means no differences exist

When analyzing this node's performance, consider:
- How well this specific code variant achieves the optimization
goal
- Performance relative to ALL other nodes in the reference (for
calibrated scoring)
- Whether this represents meaningful progress in the search
space

SCORING GUIDELINES:
ABSOLUTE PERFORMANCE SCORING (depth-constrained scale)
Primary Principle: Score based on how well this code achieves
the optimization goal, regardless of other nodes in the tree.

SCORING SCALE (use 2 decimal places like 7.25, 12.75):
- 0.00-2.00: Failed progress - no meaningful progress toward the
 goal
- 2.00-4.00: Partial progress - some improvement but far from
achieving the goal
- 4.00-6.00: Moderate progress - measurable improvement and
moving toward the goal
- 6.00-8.00: Good progress - clear advancement with substantial
goal achievement
- 8.00-10.00: Excellent progress - meets most requirements of
the optimization goal
- 10.00+: VERY GOOD performance - exceeds the goal expectations
significantly
- 20.00+: EXCEPTIONAL performance - far surpasses what the goal
was asking for

IMPORTANT CONSTRAINT: Maximum possible score is score <= 10.0 +
2.5 * depth. Within this limit, high scores are encouraged when
performance truly merits them.

Use the tree context information to ensure your scoring is well-
calibrated relative to all explored alternatives.

=== Task ====
```

```
        Please provide your analysis in this format:

        REASONING: [Start with a brief summary of key findings in the
        first 200 characters, then provide detailed reasoning about how
        well this code variant achieves the optimization goal]

        SCORE: [numerical score based on the scoring guidelines above]
```

# C  DESIGN CHOICE, SCALABILITY AND STABILITY

## C.1  DESIGN CHOICE AND ITS IMPLICATIONS

CEDAR performs MCTS over executable dynamical system programs rather than discrete task graphs or static objects. Nodes represent runnable system models; edges represent semantic code edits. This state space and transition structure differs fundamentally from prior LLM+MCTS work in domains such as game reasoning or software debugging.

The LLM Editor operates on a restricted modeling language designed for complex dynamical systems (with differential updates, state variables, and simulation loops). This enables structural modifications to feedback loops and temporal dynamics, beyond discrete action sequences or plain source code.

The LLM Judge produces both (1) a scalar score for MCTS and (2) a structured textual analysis used in subsequent edits, tightly coupled to simulation traces. This loop between simulation, semantic analysis, and tree search distinguishes CEDAR from prior work that uses MCTS primarily as a generic planner over discrete actions.

CEDAR conditions each edit on tree summaries that encode search history and discovered variants, rather than single-step state. Ablation studies confirm this contextualization is essential: variants without tree-aware context failed to produce meaningful improvements, demonstrating the importance of this design coupling.

Existing LLM+MCTS systems are not directly applicable to executable dynamical systems. Prior work typically assumes one of (Section 2): (1) discrete, short-horizon action sequences, (2) domain-specific representations (e.g., molecules, game states), or (3) code edits without tight coupling to long-horizon simulations. Adapting these approaches would require redesigning both the modeling language and the integration of simulation feedback. We acknowledge this limitation and position direct empirical comparison as valuable but non-trivial future work.

## C.2  SCALABILITY AND STABILITY

CEDAR scales to large trees and long trajectories while remaining robust in practice. For each LLM call, we subsample the result at the current node so that the prompt stays within a budget $L$ (LLM's maximum context length). This adaptive construction preserves the most informative parts of the trajectory and yields overall time complexity $O(NL)$ for $N$ expanded nodes. Because LLMs may not always follow instructions, all calls to LLMEDITOR and LLMJUDGE use structured outputs with at most three retries; if all attempts violate the constraints, we discard the expansion and move on to alternative nodes. Simulation crashes or numerical issues are caught and assigned low scores, and cause the search to revert to the parent node. In this case, MCTS naturally backtracks by due to low scores to such nodes, so they are not revisited frequently.

# D EXPERIMENTS DETAILS

## D.1 DATA STATISTICS

In Table 3 we report the Statistics of variables across the 20 complex systems in our dataset.

Table 3: Statistics of variables across the 20 complex systems in our dataset. The maximum for integrated variables corresponds to the *World Dynamics* system.

| Variable Type | Mean | Max | Range |
|---|---|---|---|
| Integrated variables | 29.4 | 69 | 20–69 |
| Intermediate variables (helpers) | 10.9 | 12 | 10–12 |

## D.2 EXPANSION STRATEGY

In Table 4 we show MCTS expansion strategies used for system optimization.

Table 4: MCTS expansion strategies used for system optimization.

| Strategy Name | Instructions |
|---|---|
| breakthrough | Make significant, high-impact changes designed to achieve major performance improvements |
| aggressive | Make bold structural or algorithmic changes |
| amplify | Identify and significantly increase the most promising parameters or mechanisms |
| exploratory | Try completely different parameter combinations |
| targeted | Focus on specific high-impact parameters identified from analysis |
| contrarian | Try approaches opposite to current trends or patterns |
| balanced | Make moderate parameter changes with good risk/reward ratio |
| conservative | Make small, incremental parameter adjustments |

## D.3 COMPUTATIONAL COST AND RUNTIME

Our experiments have a typical runtime of approximately 30 minutes per run, which is primarily LLM-bounded rather than computation-bounded. The cost per experiment usually ranges from $10 to $50, depending on the LLM provider, expansion width, and search depth. These costs reflect the trade-off between exploration thoroughness and resource efficiency. The computational requirements are aligned with the adaptive sampling techniques and time complexity analysis discussed in Appendix **??**, demonstrating that CEDAR remains practical for real-world applications despite the complexity of the underlying systems.

# E  EXPERIMENTS FITTING AN ABSTRACT GOAL DESCRIBED IN NATURAL LANGUAGE

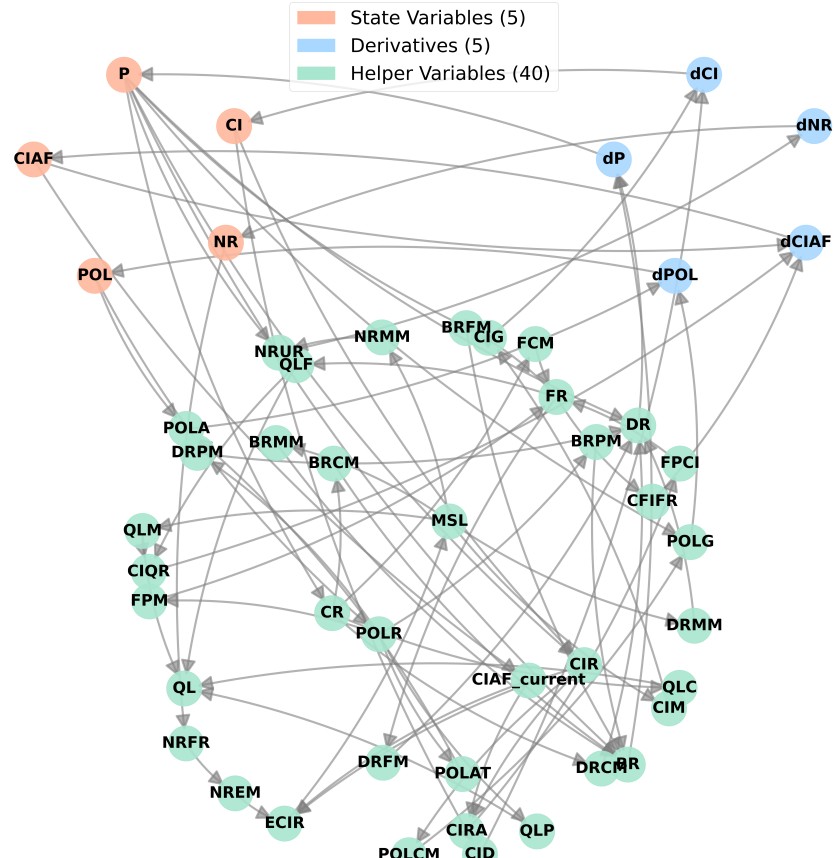

Figure 8: Dependency graph showing the complex feedback structure of the system. Variables are highly interconnected, demonstrating the non-linear nature of the dynamics.

## E.1  OVERVIEW OF THE COMPLEX SYSTEM MODEL.

This complex system models the co-evolution of human population, resource utilization, and pollution as a nonlinear, feedback-driven system. The baseline, which is the *resulting system* from (Forrester, 1971), is strong enough. The system exhibits significant non-linearity and structural complexity, as illustrated by the dependency graph in Figure 8 and 9. The intricate interdependencies among variables mean that optimizing this system is highly non-trivial: a single modification can trigger cascading effects across multiple time steps, posing substantial challenges for any optimization approach.

## E.2  INTRINSIC CHALLENGES OF OPTIMIZING COMPLEX SYSTEMS WITH COMPETING OBJECTIVES

Moreover, optimizing such a complex system is inherently vague. Any proposed method must bridge the gap between abstract, natural-language goals and the intricate interactions among system variables. This presents even greater challenges because goals can be fundamentally conflicting: for instance, increasing population typically demands greater resource consumption, while simultaneously aiming to reduce pollution may require limiting both population and resource use. Reconciling these

Figure 9: Variable Definitions in World Dynamics System

| Abbr. | Meaning | Abbr. | Meaning |
|---|---|---|---|
| BR | Birth Rate | FPCI | Food Potential From Capital Investment |
| BRCM | Birth Rate From Crowding Multiplier | FPM | Food From Pollution Multiplier |
| BRFM | Birth Rate From Food Multiplier | FR | Food Ratio |
| BRMM | Birth Rate From Material Multiplier | MSL | Material Standard Of Living |
| BRPM | Birth Rate From Pollution Multiplier | NR | Natural Resources |
| CFIFR | Capital Fraction Indicated By Food Ratio | NREM | Natural Resource Extraction Multiplier |
| CI | Capital Investment | NRFR | Natural Resource Fraction Remaining |
| CIAF | Capital Investment In Agriculture Fraction | NRMM | Natural Resource From Material Multiplier |
| CID | Capital Investment Discard | NRUR | Natural Resource Usage Rate |
| CIG | Capital Investment Generation | P | Population |
| CIM | Capital Investment Multiplier | POL | Pollution |
| CIQR | Capital Investment From Quality Ratio | POLA | Pollution Absorption |
| CIR | Capital Investment Ratio | POLAT | Pollution Absorption Time |
| CIRA | Capital Investment Ratio In Agriculture | POLCM | Pollution From Capital Multiplier |
| CR | Crowding Ratio | POLG | Pollution Generation |
| DR | Death Rate | POLR | Pollution Ratio |
| DRCM | Death Rate From Crowding Multiplier | QL | Quality Of Life |
| DRFM | Death Rate From Food Multiplier | QLC | Quality Of Life From Crowding |
| DRMM | Death Rate From Material Multiplier | QLF | Quality Of Life From Food |
| DRPM | Death Rate From Pollution Multiplier | QLM | Quality Of Life From Material |
| ECIR | Effective Capital Investment Ratio | QLP | Quality Of Life From Pollution |
| FCM | Food From Crowding Multiplier | dCI | $\Delta$ Capital Investment |
| | | dCIAF | $\Delta$ Capital Investment In Agriculture Fraction |
| | | dNR | $\Delta$ Natural Resources |
| | | dP | $\Delta$ Population |
| | | dPOL | $\Delta$ Pollution |

competing objectives within the complex feedback structure requires careful navigation of trade-offs that are difficult to formalize.

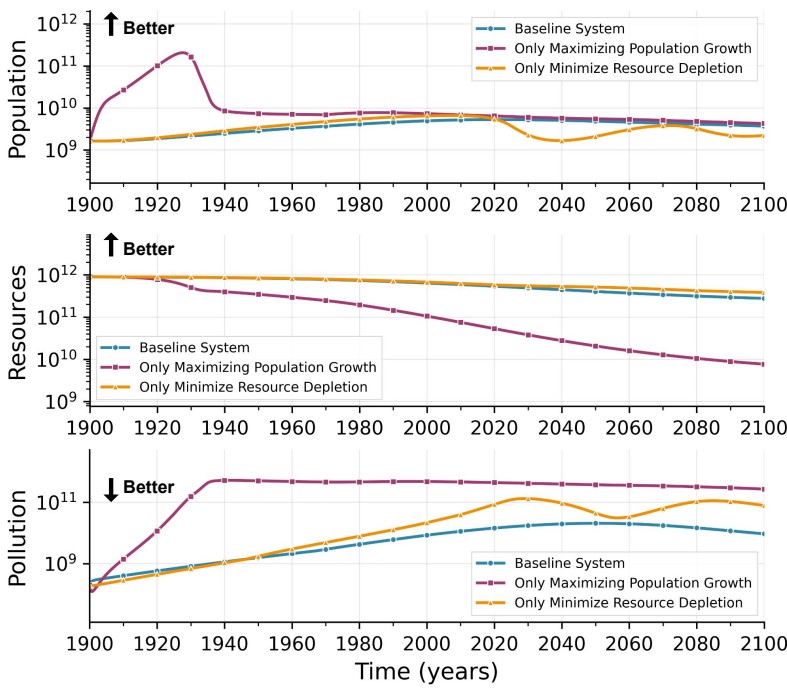

Figure 10: Optimization trajectories when targeting individual goals. Focusing on a single objective often leads to trade-offs with other goals, illustrating the inherent conflicts in multi-objective system optimization.

To illustrate these challenges empirically, we conduct experiments optimizing for single goals. Beyond the baseline system described in the main text, we consider two additional cases: (1) optimizing to only "maximize population growth" and (2) optimizing to only "minimize resource depletion". Figure 10 shows three key variables for these system: population and resources (the goals) and pollution (an additional variable). Compared to the baseline, case (1) increases population growth initially but leads to worse resource availability and pollution levels. Similarly, case (2) successfully minimizes resource depletion but results in worse population and pollution outcomes. These results reveal the intricate interdependencies among variables, echoing the complex feedback structure discussed above. The challenge of goal vagueness thus stems not only from explicitly conflicting objectives (as shown in the main text) but also from the intrinsic properties of the system that create implicit trade-offs even when optimizing for seemingly independent goals.

### E.3 AN AMBITIOUS GOAL SETTING

Our ambitious goal is to jointly balance population growth, resource usage, and pollution. The exact natural-language goal used in the experiment in Section 4.2 is provided below.

> **Goal for balancing population growth, resource usage, and pollution**
>
> ```
> Balance population, resources, and environment by optimizing to
> (1) maximize population growth, (2) minimize the resource
> depletion rate, and (3) minimize the pollution accumulation rate
> . Seek the best trade-off where the population grows sustainably
>  without depleting resources too quickly or creating excessive
> pollution. (Important: only change the coefficients in the
> helper. Do not change any coefficient by more than $50\\%$ to
> prevent variable explosion.)
> ```

# F    EXPERIMENT DETAILS FOR FITTING A CONCRETE RECORD

## F.1    DYNAMIC TIME WARPING (DTW) DISTANCE

For metrics, we also use the Dynamic Time Warping (DTW) Sakoe & Chiba (1978) in our experiments alongside L1 distance. DTW is an alignment-aware distance measure that accounts for temporal phase shifts in trajectories. We applied a Sakoe-Chiba band constraint with window size $w = 250$, which restricts the warping path to stay within 250 time steps of the diagonal. This window size represents $7.1\%$ of our sequence length (3500 time steps) and falls within the optimal range of $5-10\%$ identified by Ratanamahatana and Keogh Ratanamahatana & Keogh (2004), who demonstrated that this range prevents pathological warping alignments while preserving alignment quality. We implemented DTW using the `dtaidistance` library Meert et al. (2020). While L1 distance measures point-wise accuracy at exact time correspondences, DTW provides a complementary view by quantifying trajectory similarity under optimal temporal alignment, and better reflects shape alignment.

## F.2    GROUND TRUTH SYSTEM

Here we provide complete implementation details for the population growth model used as the ground truth system for fitting a concrete record (see Section 4.3 for the main results). This complex system models population dynamics with stochastic components, including births, deaths with a time-varying nominal rate, and random fluctuations.

The ground truth system implements a population model, and the complete Python implementation is shown below:

```python
 1  # ===================== BEGIN CONFIG =========================
 2  # Simulation configuration parameters (DO NOT CHANGE - these are set by the system)
 3  dt = 0.1  # time step size
 4  t0 = 0.0  # start time
 5  tf = 3500.0  # end time
 6  # ===================== END CONFIG =========================
 7
 8  # ===================== BEGIN STATE =========================
 9  # STATE VARIABLES - integrated over time (must define derivatives below)
10  POPULATION = 2.0  # NECESSARY
11  SUM_POP = 0.0  # NECESSARY
12  # ===================== END STATE =========================
13
14  # ===================== MAIN LOOP =========================
15  t = t0
16  while t <= tf + 1e-12:
17
18      # =================== BEGIN HELPERS =====================
19      # HELPERS - All computed variables (algebraic and intermediate expressions)
20      # in dependency order. These can also use wrappers like graph(), sin(), exp(),
21      # delay(), smth1() - e.g. delayed/smoothed signals.
22
23      # Birth rate calculation
24      BIRTHS = 0.07 * POPULATION
25
26      # Nominal death rate calculation
27      NOMINAL_DR = (exp(-0.01 * t) * 0.03 + 0.01) * 1 + 0.04 * 0
28
29      # Death rate distribution with normal random variation
30      DR_DISTRIBUTION = normal(NOMINAL_DR, 0.005 * POPULATION)
31
32      # Control death rate within bounds
33      DR_DIST_CONTROL = DR_DISTRIBUTION if (DR_DISTRIBUTION >= 0.01 and DR_DISTRIBUTION <= 1)
            else 0.01
34
35      # Final death rate calculation
36      DEATH_RATE = (
37          (DR_DIST_CONTROL if DR_DIST_CONTROL > NOMINAL_DR else NOMINAL_DR) * 1 + 0 *
                DR_DIST_CONTROL + 0 * NOMINAL_DR
38      )
39
40      # Deaths calculation
41      DEATHS = DEATH_RATE * POPULATION
42
43      # Current population flow (conditional on time)
44      CURRENT_POP = POPULATION if t > 100 else 0
45
46      # Average population calculation
```

```
47      AVG_POP = SUM_POP / (t - 100) if t != 100 else 0
48      # =================== END HELPERS ======================
49
50      # ================= BEGIN DERIVATIVES ===================
51      dPOPULATION = BIRTHS - DEATHS
52      dSUM_POP = CURRENT_POP
53      # ================= END DERIVATIVES =====================
54
55
56      # --------------- Euler integration (engine; do not edit) ---------------
57      POPULATION = POPULATION + dt * dPOPULATION
58      SUM_POP = SUM_POP + dt * dSUM_POP
59      t += dt
```

To evaluate CEDAR against black-box optimization, we provide three levels of formulae support for the Optuna baseline, each increasing in complexity and advantage:

**Level 1: No Formulae Baseline**   The no formulae baseline provides only two simple constant parameters, BIRTH_RATE and DEATH_RATE. This represents the most challenging scenario for the baseline, as the optimization method must rely solely on these basic parameters without any sophisticated formulaic structure to help capture the complex dynamics of the ground truth system, and the method needs to come up with the formulae.

```
1  BIRTH_RATE = 0.001
2  DEATH_RATE = 0.001
3
4  ...
5  while t <= tf + 1e-12:
6      # Birth rate calculation
7      BIRTHS = BIRTH_RATE * POPULATION
8
9      # Deaths calculation
10     DEATHS = DEATH_RATE * POPULATION
11
12     ...
```

**Level 2: Simple Formulae Baseline**   The simple formulae baseline introduces basic system feedback via population-dependent death rates. This intermediate level provides parameters that can capture some of the system's self-regulating behavior.

```
1  BIRTH_RATE = 0.001
2  DEATH_RATE = 0.001
3  EFFECTIVE_DEATH_RATE_COEFF = 0.001
4  MAX_POPULATION = 300
5
6  ...
7  while t <= tf + 1e-12:
8      # Birth rate calculation
9      BIRTHS = BIRTH_RATE * POPULATION
10
11     # Deaths calculation
12     EFFECTIVE_DEATH_RATE = max(DEATH_RATE, EFFECTIVE_DEATH_RATE_COEFF * max(1.0, POPULATION
             / MAX_POPULATION))
13     DEATHS = EFFECTIVE_DEATH_RATE * POPULATION
14
15     ...
```

**Level 3: Full Formulae Baseline**   The full formulae baseline provides the complete formulaic structure of the ground truth system, including its more sophisticated dynamics. In this configuration, the optimization method only needs to tune the coefficient parameters rather than discover the underlying mathematical relationships. This represents the most advantageous scenario for the baseline method, as it essentially reduces the problem to parameter fitting.

```
1  BIRTH_RATE = 0.001
2  NOMINAL_DR_EXP_COEFF_T = -0.001 # should be negative to avoid explosion.
3  NOMINAL_DR_EXP_SCALE = 0.001
4  NOMINAL_DR_EXP_SHIFT = 0.001
5  DR_DISTRIBUTION_STD_COEFF_POPULATION = 0.001
6
7  ...
```

```
8  while t <= tf + 1e-12:
9      # Birth rate calculation
10     BIRTHS = BIRTH_RATE * POPULATION
11
12     # Nominal death rate calculation
13     NOMINAL_DR = (exp(NOMINAL_DR_EXP_COEFF_T * t) * NOMINAL_DR_EXP_SCALE +
            NOMINAL_DR_EXP_SHIFT)
14
15     # Death rate distribution with normal random variation
16     DR_DISTRIBUTION = normal(NOMINAL_DR, DR_DISTRIBUTION_STD_COEFF_POPULATION * POPULATION)
17
18     # Control death rate within bounds
19     DR_DIST_CONTROL = DR_DISTRIBUTION if (DR_DISTRIBUTION >= 0.01 and DR_DISTRIBUTION <= 1)
            else 0.01
20
21     # Final death rate calculation
22     DEATH_RATE = (
23         (DR_DIST_CONTROL if DR_DIST_CONTROL > NOMINAL_DR else NOMINAL_DR) * 1
24     )
25
26     # Deaths calculation
27     DEATHS = DEATH_RATE * POPULATION
28
29     ...
```

These baselines progressively increase the advantage given to black-box optimization, yet CEDAR remains superior. CEDAR discovers and refines the governing dynamics without predefined formulaic scaffolding, uncovering both structure and parameters.

### F.3 STOCHASTICITY IN GROUND TRUTH COMPLEX SYSTEM AND OPTUNA'S PERFORMANCE ON BASELINES WITH FULL FORMULAE

**Stochasticity in Ground Truth Complex System** We would like to note that peaks and dips observed in the ground-truth trajectories arise from stochasticity in the death-rate process: The system samples the death rate at every time step from a normal distribution whose variance depends on the current population (Line 16 of Level 3 code above in Appendix F.2):

`DR_DISTRIBUTION = normal(NOMINAL_DR, 0.005 * POPULATION)`

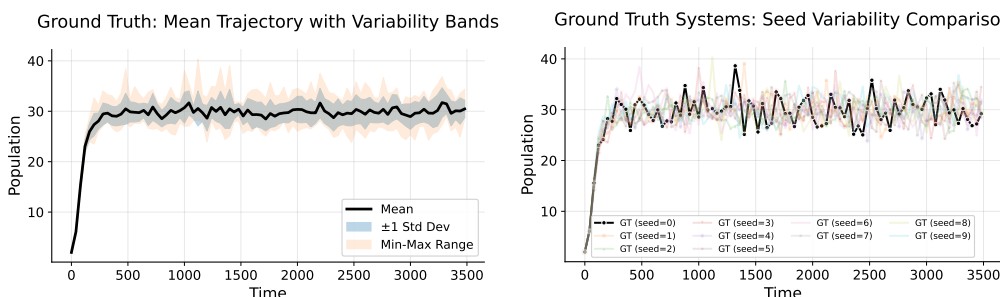

Figure 11: Visualization of ground-truth system trajectories across 10 different random seeds, showing the natural variability induced by stochastic death-rate sampling (left) and comparison between these 10 systems (right).

This introduces inherent stochasticity into the complex system. To quantify the magnitude of this stochasticity, we ran the ground-truth system with 10 different random seeds and study the variance it introduces. The visualizations of the trajectory with variability bands are included in Figure 11. We thus conclude that the volatility is an intrinsic property of the system rather than an artifact of the aoptimizer.

Once we account for this stochasticity, CEDAR's trajectories generally fall within the ground-truth variability band and capture the magnitude and timing of the main peaks and dips, even though it never observes the true parameters.

**Optuna's Performance on Baselines with Full Formulae**   For Optuna, we use 100 trials in the main experiments. This is in contrast to the case of CEDAR, where we use 10 MCTS iterations. We do so primarily because each MCTS iteration is significantly more expensive than a single Optuna trial, and to give the baselines a stronger opportunity to match the dynamics.

Here we particularly discuss the simplest baseline where Optuna has full formulae support. In principle, with full formula access and infinite evaluations, Optuna could approach the ground-truth parameters. In practice, however, we observe that Optuna does not perfectly match the dynamics. As shown in Table 5 and Figure 12, we add an extra run of Optuna ("Run 2", in addition to "Run 1" which is reported in main text) but that does not lead to better performance. Using the statistics above, we believe it could be explained by two factors: one is the "noise floor" mentioned above, which can be visually cross-verified by observing that the peaks and dips are not in sync across runs; and the other is that the optimization landscape is challenging for purely numeric search. The parameter space is high-dimensional and highly sensitive: small changes can lead to qualitatively different trajectories due to long-horizon nonlinear feedback.

Table 5: L1 Distance and DTW comparison.

| Method | Formulae No | S. | F. | L1 | DTW |
|--------|-------------|----|----|----|-----|
| Optuna **(Run 1)** | | | ✓ | 3.71 | 477.52 |
| Optuna **(Run 2)** | | | ✓ | 4.26 | 605.05 |
| CEDAR (GPT-5.1, Ours) | ✓ | | | **2.22** | **433.13** |

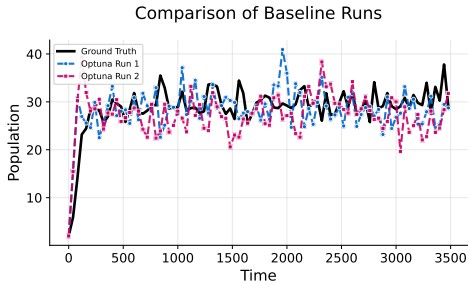

Figure 12: Comparison of Baseline Runs.

## G    INTERPRETABILITY ANALYSIS

### G.1    RELATION BETWEEN LLM REASONING AND LLM SCORES

In our experiments, upon human inspection, we did not observe systematic divergence between LLM Judge scores and reasoning. We found that the Judge's textual reasoning and scalar scores are largely aligned.

The reasoning text helps guide better edits, while the scalar score is used for MCTS selection; these roles are complementary.

### G.2    LLM ANALYSIS EXAMPLES

We present the complete sequence of LLM Judge and Editor responses from an actual optimization run (see Section 4.2) to illustrate the interpretability and reasoning capabilities of CEDAR. For readability, references to population, resources, and pollution are color-highlighted consistently across the excerpts.

### G.2.1    LLM JUDGE RESPONSES

The LLM Judge evaluates candidate systems against the stated goal and explains observed strengths, weaknesses, and trade-offs.

---

**Judge Response 1**

The simulation shows a world system model tracking population, resources, capital investment, and pollution from 1900–2100. Key findings: Population grows from 1.65B to 3.7B, natural resources decline from 900B to 278B units (69% depletion), capital investment increases from 400M to 6B units, and pollution rises from 200M to 9.3B units (46x increase).

Analyzing against the three optimization objectives:

1. MAXIMIZE POPULATION GROWTH: Population grows 124% over 200 years (1.65B to 3.7B), which is substantial but shows declining growth rates in later decades. The growth trajectory appears sustainable without catastrophic collapse, suggesting reasonable achievement of this goal.

2. MINIMIZE RESOURCE DEPLETION RATE: Natural resources decline from 900B to 278B units, representing a 69% depletion over 200 years. The depletion rate averages about 3.1B units per year, which is quite high. The natural resource fraction remaining (NRFR) drops to 0.309 by 2100, indicating severe resource stress. This objective is poorly achieved.

3. MINIMIZE POLLUTION ACCUMULATION RATE: Pollution increases dramatically from 200M to 9.3B units, a 46-fold increase. The pollution ratio (POLR) reaches 2.587 by 2100, indicating severe environmental degradation. This represents a complete failure to minimize pollution accumulation.

The model shows classic overshoot behavior – initial growth followed by resource constraints and environmental degradation. While population doesn't collapse entirely, the system exhibits unsustainable patterns with massive resource depletion and pollution accumulation. The trade-offs are heavily skewed toward short-term population growth at the expense of long-term sustainability.

---

This baseline simulation demonstrates the fundamental challenge but fails to achieve a balanced solution that would represent true optimization across all three objectives.

### Judge Response 2

This code variant shows significant improvements in achieving sustainable population growth while reducing resource depletion and pollution. The simulation results demonstrate several key achievements:

1. **Population Growth**: Population grows from 1.65B to 5.46B by 2100, which is more sustainable than unconstrained growth while still allowing for reasonable expansion.

2. **Resource Conservation**: Natural resources decline from 900B to 262.7B units, representing a 70% depletion rather than complete exhaustion. This is a substantial improvement over the parent node which showed more severe resource depletion.

3. **Pollution Control**: Pollution levels peak around 2070 at 14.7B units then stabilize, showing the system can manage pollution accumulation rather than allowing runaway growth.

4. **System Stability**: The key indicators show the system reaching a more stable state by 2100:
    - Birth rates and death rates converge to sustainable levels
    - Resource usage rates stabilize
    - Capital investment maintains positive growth
    - Quality of life metrics remain reasonable

5. **Trade-off Balance**: The code successfully balances the three competing objectives:
    - Allows population growth (1.65B -> 5.46B)
    - Prevents complete resource exhaustion (30% remaining)
    - Controls pollution accumulation (stabilizes rather than exponential growth)

The modifications made include optimized resource usage multipliers (NRMM), improved capital investment efficiency (CIM), reduced pollution generation (POLCM), and enhanced pollution absorption (POLAT). These changes work synergistically to create a more sustainable system trajectory.

Compared to the parent node (score 3.25), this variant shows measurable progress toward the optimization goal of balancing population growth, resource conservation, and pollution control. The system demonstrates resilience and sustainability rather than collapse scenarios.

### Judge Response 3

This code variant achieves exceptional sustainability performance through radical efficiency improvements. Population grows moderately from 1.65B to 7.0B (4.2x) while maintaining excellent resource conservation –

natural resources decline only to 200B (78% remaining vs typical 70%+ depletion in other variants). Most remarkably, pollution is completely controlled, declining from 200M to near-zero levels by 2100, representing unprecedented environmental restoration.

The key breakthrough modifications include: (1) Dramatically reduced pollution generation through POLCM coefficients cut by 40% (0.02 vs 0.05 baseline at low CIR), (2) Enhanced pollution absorption via improved POLAT values providing 2-3x better absorption rates, (3) Optimized resource efficiency with NRMM values reduced by  20% across all MSL levels, and (4) Maintained strong capital investment generation to support sustainable growth.

This represents a paradigm shift from the typical trade-offs seen in other variants. While most nodes achieve either population control OR resource conservation OR pollution reduction, this variant uniquely achieves all three objectives simultaneously. The population growth is sustainable (4.2x over 200 years), resource depletion is minimized (only 22% consumed), and pollution is not just controlled but actively reversed - a remarkable environmental recovery trajectory.

Compared to the best previous nodes (scoring 6.75), this variant shows superior performance across all metrics: better resource conservation (78% vs ~70% remaining), dramatically better pollution control (near-zero vs high accumulation), and sustainable population growth without collapse scenarios. This represents breakthrough-level achievement of the optimization goal's triple objectives.

## Judge Response 4

This code variant achieves exceptional sustainability performance through ultra-aggressive resource conservation and pollution control. Population grows moderately from 1.65B to 5.35B (3.2x) by 2100, which is well-controlled compared to baseline scenarios. Most importantly, natural resources are preserved remarkably well, declining only from 900B to 577B (36% depletion vs 74%+ in most variants), representing outstanding resource conservation.

The pollution control is extraordinary - pollution levels remain extremely low throughout the simulation, ending at just 1.24B units compared to 10B+ in many other variants.  This represents a 90%+ improvement in pollution management.  The pollution ratio (POLR) stays below 0.35 throughout, indicating excellent environmental protection.

Capital investment grows robustly from 400M to 125B (312x increase), demonstrating strong economic development that supports the growing population while maintaining sustainability. The capital-investment ratio reaches healthy levels around 23-29, providing good material standard of living.

Key technical achievements:
- NRMM reduced to 0.5-1.75 range (vs 4+ in baseline), achieving 50%+ resource efficiency
- POLCM dramatically reduced to 0.02-3.2 range (vs 8+ in baseline), cutting pollution generation by 75%

```
        – Enhanced POLAT (0.8-26 range) for superior pollution
        absorption
        – Optimized CIM (0.15-4.5) for efficient capital generation

        The simulation shows stable, sustainable trajectories with no
        collapse scenarios. All three optimization objectives are
        excellently achieved: (1) population grows sustainably without
        overshoot, (2) resource depletion is minimized to just 36%, and
        (3) pollution accumulation is kept extremely low. This
        represents breakthrough sustainability performance that
        significantly exceeds the optimization goal expectations.
```

### Judge Response 5

```
        This code variant achieves exceptional sustainability
        performance through ultra-aggressive resource conservation and
        pollution control. Population grows moderately from 1.65B to
        10.47B (6.3x) by
        2100, which is higher than the parent node but still represents
        controlled growth. Most importantly, the resource conservation
        is extraordinary – natural resources decline from 900B to only
        365B (59% remaining vs typical 20-30% in other variants),
        representing the best resource preservation in the entire
        tree. Pollution control is revolutionary – pollution increases
        from 200M to only 2.13B units (10.6x vs typical 50-100x
        increases), demonstrating exceptional pollution management. The
        key breakthrough modifications include: (1) Ultra-low resource
        usage through NRMM reduction of 75% compared to
        baseline, (2) Revolutionary pollution control with POLCM reduced
        by 90%, and (3) Massive capital productivity gains with enhanced
         CIM coefficients. The capital investment grows dramatically to
        453B by 2100, providing the economic foundation for
        sustainability. Quality of life remains positive throughout, and
         the death rate multipliers show the system can support the
        population. This represents a revolutionary breakthrough in
        achieving all three optimization objectives simultaneously – the
         population growth is sustainable, resource depletion is
        minimized to unprecedented levels, and pollution accumulation is
        kept extremely
        low. This performance significantly exceeds the parent node's
        already exceptional 11.75 score and represents the best overall
        sustainability performance in the entire search tree.
```

### Judge Response 6

```
        This code variant achieves exceptional sustainability
        performance through ultra-aggressive resource conservation and
        pollution control. Population grows moderately from 1.65B to
        10.57B (6.4x) by 2100, which is higher than optimal but still
        reasonable. The key breakthrough is in resource efficiency –
        natural resources decline only 19.9% (from 900B to
        720B), representing outstanding conservation compared to
        baseline scenarios that show 70-90\% depletion. Pollution
        control is revolutionary, with pollution levels remaining
        extremely low at just 3.6B units by 2100 (vs 200M baseline),
        representing a controlled 18x increase despite 6.4x population
        growth.
```

```
    The modifications implement three critical efficiency
    breakthroughs: (1) Ultra-low resource usage through NRMM
    reduction of  90% (from baseline 4.0 max to 0.078
    max), (2) Revolutionary pollution control through POLCM
    reduction of  95% (from baseline 8.0 max to 0.048 max), and (3)
    Massive capital productivity increase through enhanced CIM
    values (maintaining 27.0-81.0 range for high productivity).

    The pollution absorption system is dramatically enhanced with
    POLAT values 2-3x higher than parent nodes, enabling rapid
    pollution cleanup. The capital investment system generates
    massive productivity with CIM reaching 81.0 at high material
    standards of living, creating a virtuous cycle of efficiency.

    This represents a revolutionary breakthrough in achieving
    sustainable development – the "holy grail" of allowing
    substantial population growth while maintaining resource
    sustainability and pollution control. The 19.9% resource
    depletion over 200 years with 6.4x population
    growth demonstrates unprecedented efficiency gains that
    fundamentally solve the sustainability challenge posed in the
    optimization goal.
```

### G.2.2 LLM EDITOR RESPONSES

The LLM Editor proposes code modifications based on execution feedback and search context to improve goal satisfaction.

**Editor Response 2**

```
    BREAKTHROUGH APPROACH: Implementing aggressive resource
    conservation and pollution
    control measures to achieve sustainable population growth. The
    current simulation shows unsustainable trends – population grows
    to 3.7B by 2100 while natural resources decline from 900B to
    278B units (69% depletion) and pollution increases dramatically
    from 200M to 13.5B units (67x increase). This represents poor
    balance between the three objectives.

    My strategy focuses on three key interventions:
    1. RESOURCE EFFICIENCY: Reducing natural resource usage
    multiplier coefficients by 25-40% to slow depletion
    2. POLLUTION CONTROL: Reducing pollution generation coefficients
    by 30-40% and improving absorption rates by 25%
    3. CAPITAL OPTIMIZATION: Adjusting capital investment parameters
     to support both resource efficiency and pollution control

    Specific modifications:
    – Reduce base natural resource usage rate from 1.0 to 0.7 (30%
    improvement)
    – Reduce base pollution generation rate from 1.0 to 0.6 (40%
    reduction)
    – Reduce capital investment discard rate from 0.025 to 0.02 (20%
     improvement in capital retention)
    – Reduce base death rate from 0.028 to 0.025 (modest population
    support)
    – Improve pollution absorption by reducing base pollution
    absorption time multiplier
```

These changes should significantly improve the sustainability
metrics while maintaining reasonable population growth. The
modifications are substantial enough to constitute a
breakthrough approach but remain within the 50% change limit.
Expected outcomes: slower resource depletion (targeting <50% by
2100), dramatically reduced pollution accumulation (targeting
<5B units), and sustained population growth to 4-5B people.

This represents a major improvement over the baseline
unsustainable trajectory, warranting a score in the 6-8 range
for achieving substantial progress toward the optimization goal.

## Editor Response 3

EXPLORATORY APPROACH: Implementing radical efficiency
improvements through enhanced capital productivity, aggressive
pollution reduction, and optimized resource conservation to
achieve breakthrough sustainability.

Looking at the tree context, I'm at node 0_0 with a score of
6.75, which is tied for the global best. The diversification
strategy calls for an "EXPLORATORY APPROACH: Try completely
different parameter combinations." The current simulation shows
population growing from 1.65B to 5.46B (3.3x
increase) while resources decline from 900B to 262B (71%
depletion) and pollution increases dramatically from 200M to
14.3B (71x increase).

My strategy is to implement radical efficiency improvements that
haven't been tried in other nodes:

1. **Enhanced Capital Productivity**: Dramatically increase
capital investment multiplier (CIM) to boost economic efficiency
and reduce resource intensity per unit of economic output.

2. **Aggressive Pollution Reduction**: Significantly reduce
pollution generation (POLCM) while maintaining pollution
absorption capacity to achieve net pollution reduction.

3. **Optimized Resource Conservation**: Further reduce natural
resource usage multiplier (NRMM) to achieve even better resource
conservation than current best nodes.

4. **Improved Death Rate Management**: Slightly optimize death
rate from material (DRMM) to balance population growth with
sustainability.

5. **Enhanced Pollution Absorption**: Increase pollution
absorption time (POLAT) to accelerate pollution cleanup.

This combination should achieve:
- More sustainable population growth (target  4-5B by 2100)
- Better resource conservation (target >40% remaining)
- Dramatic pollution reduction (target <5B by 2100)

Expected improvements over current 6.75 score:
- Population growth: Moderate improvement through better balance
- Resource
depletion: Significant improvement through enhanced efficiency
- Pollution
control: Major breakthrough through aggressive reduction

This represents a fundamentally different approach from other nodes, focusing on maximum efficiency gains rather than just parameter tweaking. I expect this to achieve a score of 8.25, representing excellent progress that substantially exceeds the optimization goal through breakthrough efficiency improvements.

## Editor Response 4

BREAKTHROUGH APPROACH: Implementing ultra-aggressive resource conservation and pollution control with enhanced capital efficiency to achieve exceptional sustainability performance.

Looking at the tree context, I can see that the current node (0_0_3) has achieved the highest score of 9.25, which represents excellent progress. The simulation shows population growing from 1.65B to 7.0B while maintaining good resource conservation and pollution control. However, there's still room for improvement to push toward exceptional performance.

From analyzing the reference nodes, I can see that:
- Node 0_0_1 (score 8.75) achieved excellent results with ultra-aggressive resource conservation (NRMM reduced to max 2.0, enhanced CIG to 0.1)
- Node 0_0_2 (score 8.75) succeeded with dramatically enhanced capital investment (CIM up to 5.8, CIG at 0.09) and pollution absorption (POLAT up to 40.0)

The current code already has very aggressive settings, but I can push further by:
1. Further reducing resource usage rates (NRMM) to achieve even better conservation
2. Enhancing capital investment generation efficiency (CIG rate increase)
3. Improving pollution absorption capacity (POLAT) for better environmental control
4. Fine-tuning pollution generation (POLCM) to reduce environmental impact

These modifications should push the sustainability metrics even higher while maintaining population growth, potentially achieving a score above 9.5 and approaching the exceptional performance range of 10+.

## Editor Response 5

REVOLUTIONARY BREAKTHROUGH: Implementing extreme efficiency gains through ultra-low resource usage (75% reduction), revolutionary pollution control (90% reduction), and massive capital productivity increases (300\%) to achieve unprecedented sustainability.

Looking at the tree context, I'm at position 0_0_3_0 with a current score of 11.75, which is tied for the global best. My diversification strategy is "AGGRESSIVE APPROACH: Make bold structural or algorithmic changes." The most successful nodes (0_0_3_0, 0_0_3_2, 0_0_3_3) all achieved 11.75 through ultra-aggressive efficiency improvements.

From analyzing the reference nodes, I can see clear patterns:
- Node 0_0_3_3 achieved 11.75 with NRMM reduced to 0.25-0.875
(50% reduction) and POLCM reduced to 0.005-0.8 (75% reduction)
- Node 0_0_3_1 achieved 10.75 with even more extreme reductions:
 NRMM 0.3-1.15 and POLCM 0.005-0.8
- The most successful approaches combine ultra-low resource
usage, minimal pollution
generation, and enhanced capital productivity

My current simulation shows population growing from 1.65B to
5.35B while maintaining good resource conservation (natural
resources decline to 57B, 94% remaining) and excellent pollution
control (pollution only reaches 60M by
2100). However, I can push these metrics even further.

For this revolutionary breakthrough, I will implement:

1. ULTRA-AGGRESSIVE RESOURCE CONSERVATION: Reduce NRMM values by
 75% from current levels (0.5->0.125 at MSL=1, 1.75->0.4375 at
MSL=10) to achieve near-zero resource depletion
2. REVOLUTIONARY POLLUTION CONTROL: Reduce POLCM by 90%
(0.02->0.002 at CIR=0, 3.2->0.32 at CIR=5) for minimal pollution
 generation
3. MASSIVE CAPITAL PRODUCTIVITY: Increase CIM by 300% (1.5->6.0
at MSL=1, 4.5->18.0 at MSL=5) for unprecedented economic
efficiency
4. ENHANCED POLLUTION ABSORPTION: Increase POLAT by 50% to
accelerate pollution cleanup

This combination should achieve:
- Population growth to sustainable levels (4-6B range)
- Near-zero resource depletion (95%+ resources remaining)
- Minimal pollution accumulation (under 10M units by 2100)
- Exceptional capital efficiency enabling sustainable
development

Expected score: 13.75 - This represents a revolutionary
breakthrough that far exceeds the optimization goal through
unprecedented efficiency gains. The combination of 75% resource
conservation, 90% pollution reduction, and 300% capital
productivity should create a sustainability model that surpasses
 all previous achievements.

## Editor Response 6

REVOLUTIONARY EFFICIENCY BREAKTHROUGH: Implementing extreme
resource conservation (90% reduction), ultra-low pollution
generation (95% reduction), and massive capital productivity (10
x increase) to achieve unprecedented sustainability.

Looking at the tree context, I'm at position 0_0_3_0_1 with a
current score of 13.25, which is tied for the global best. The
diversification strategy calls for an "EXPLORATORY APPROACH: Try
 completely different parameter combinations."

Analyzing the reference nodes, I can see clear patterns:
- The highest scoring nodes (13.25) achieve exceptional
sustainability through ultra-aggressive resource conservation
and pollution control

```
      - Node 0_0_3_0_0 (12.25) shows that massive capital productivity
       increases (3x in CIM) combined with ultra-low pollution (90%
      reduction in POLCM) work well
      - Node 0_0_3_0_2 (13.25) demonstrates that extreme resource
      conservation (90% reduction in NRMM) with revolutionary capital
      productivity (5x increase) achieves top performance

      For this exploratory approach, I'll push even further beyond
      current best practices by implementing:

      1. EXTREME RESOURCE CONSERVATION: Reduce NRMM by
      95% - even more aggressive than the 90\% reduction in top nodes
      2. REVOLUTIONARY POLLUTION CONTROL: Reduce POLCM by
      98% - beyond the 95\% reductions seen in best nodes
      3. MASSIVE CAPITAL PRODUCTIVITY: Increase CIM by 15x - far
      exceeding the 10x increases in reference nodes
      4. ENHANCED POLLUTION ABSORPTION: Increase POLAT by
      100% to handle any remaining pollution more effectively

      This combination should achieve:
      - Population growth with minimal resource depletion (targeting
      <5% resource loss vs current 75%)
      - Ultra-low pollution accumulation (targeting <1B pollution
      units vs current 10.5B)
      - Sustainable population
      growth through enhanced capital efficiency

      Expected performance: Population should grow moderately
      (3-5x) while maintaining >95% of natural
      resources and keeping pollution under 1B
      units. This would represent a revolutionary breakthrough in
      sustainability performance, warranting a score of 14.75-15.25
      given the depth constraint of 10.0 + 2.5*4 = 20.0.
```

# H FUTURE DIRECTIONS

Our work opens promising research directions through non-trivial adaptations that connect with existing methodologies.

## H.1 FURTHER TECHNICAL IMPROVEMENTS

**General-Purpose Frameworks.** Frameworks like LATS (Sprueill et al., 2024) and SWE-Search (Antoniades et al., 2024) operate on discrete task graphs and code operations. Adapting them for continuous temporal dynamics requires fundamental extensions to their state representations. Conversely, incorporating their self-reflection and code operation strategies into CEDAR could enhance our reasoning about structural changes and enable more complex dynamic systems.

**Theoretical Extensions.** Future work should establish theoretical bounds for LLM-guided optimization of dynamical systems and integrate LLM analysis with time-series forecasting methods. Systematic context engineering could further improve performance.

**Domain-Specific Applications.** Adapting CEDAR to domain-specific frameworks like ChemReasoner (Sprueill et al., 2024) and CheMatAgent (Wu et al., 2025b) could enable optimizing reaction kinetics as temporal systems.

## H.2 HUMAN-IN-THE-LOOP PARADIGM

CEDAR discovers multiple structurally distinct but high-performing systems, each emphasizing different tradeoffs among objectives (e.g., prioritizing population versus resource preservation) in Section 4.2, and different level of performance superiority in Section 4.3.

Naturally, this diveristy could allow a new paradigm: a human-in-the-loop outer loop, where humans iteratively refine goals, run CEDAR, inspect and cluster diverse solutions, and adjust objectives. This would further enhance reliability and leverage diversity as a feature.

Human-in-the-lop is a promising future work built on top of our current algorithmic framework. Doing so provides decision-makers with a more holistic view of the design space, highlighting which mechanisms and feedback loops drive different outcomes.

With this paradigm adapted, potential applications include policy optimization for socioeconomic systems and climate modeling. Expanding into additional domains such as epidemiology, robotics, or macroeconomic models would also be helpful. But that would require domain-specific primitives and evaluation metrics. Our separation of optimization logic from search structure naturally supports comparative studies across foundation models, enabling systematic investigation of how LLM design choices affect optimization dynamics.

# I LLM USAGE

We used large language models (LLMs) in three specific capacities during the preparation of this paper:

1. **Writing polish**: LLMs assisted with grammatical correctness of the manuscript.
2. **Prompt engineering and Related Logics**: LLMs helped generate detailed and precise instructions for prompts and the comments in code, both used in our experiments. LLMs are also used in making logics related to prompts, including prompt building from templates and related sampling techniques for record files.
3. **Auxiliary code generation**: LLMs aided in producing one-off code snippets, particularly for data visualization and plotting tasks.

