# OpenReview forum: "CEDAR: Agent‑Orchestrated Tree Search for Goal‑Directed Optimization of Complex Systems"
_ICLR.cc/2026/Conference — Submitted to ICLR 2026_

### Official Review · Reviewer_rRnD · 2025-10-27

**Soundness:** 2
**Presentation:** 3
**Contribution:** 2
**Rating:** 2
**Confidence:** 3

**Summary:**

This paper introduces CEDAR, a LLM + MCTS–inspired procedure to iteratively generate models of complex dynamical systems toward satisfying (potentially vague) user-defined goals. The method is well described, there are several experimental results, and the work seems largely reproducible. However, to meet the acceptance bar, the theoretical justifications should be further developed, as detailed in the weaknesses and questions section below.

**Strengths:**

- The motivation of the work is important: Optimising complex systems under natural language goals

- The background and related work section covers the contemporary work

- The presentation throughout the paper is clear, with good diagrams and visualisations

- There are several included experiments covering optimising (under vague goals and direct ground truth) and interpretability.

**Weaknesses:**

The main limitation of the work is the lack of theoretical depth and justifications. While the searching algorithm is clearly tree based, with selection, expansion, evaluation, and backpropagation steps, the connection to true monte carlo sampling and UCT is not clear. Does the approach inherit any of the desirable characteristics from UCT? How does using the uniform expansion strategy with different prompts relate to more classically balances between exploration and exploration?  A few of these decisions seem quite adhoc, and discussion of alternatives and tradeoffs is warranted.

Additionally,  the key aspect of the work is on the tree search component, but it is not clear how this distinguishes itself from the other LLM + MCTS works (which are cited) besides a different application area.

These weaknesses are detailed more specifically in the questions section.

**Questions:**

**Algorithm**:
- From an algorithmic perspective, what is novel between CEDAR and the many other LLM+MCTS methods mentioned? Why are these not used as comparisons?

- What is "f" in the SCORE(v) equation?

- Why the hard threshold tau, and not a more explicit balance between explore/exploit as in traditional MCTS?

**Experiments**:

- In the comparisons, why does the baseline (with full formula) not perfectly recreate the observed dynamics? How many samples/runs are used in the baseline versus the proposed? Are the number of samples equated, and in the limit of infinite runs would the baseline converge to the true dynamics. Are you guaranteed such results with your expansion strategy?

- The L1 distance seems a relatively poor measure for system dynamics, as indicated by the lowest L1 for CEDAR but very flat dynamics not capturing any of the peaks and dips (which I imagine is seasonality?). Alternative metrics, e.g. Dynamic Time Warping should be used.

- In Sec 4.2: There are no comparisons in Figure 4. What is the baseline system here? v0? At minimum it would be good to see one optimising each goal on the others to show the tradeoff, e.g. optimise for population and plot that curve on the resources and pollution plots to show the impact.

Small suggestions, but 4.4 would be better made as a subsection of 4.2 for clarity/flow.

---

> ### Author Response · Authors · 2025-11-24
> **Responses to Reviewer rRnD (Part 1)**
>
> We thank the reviewer for the detailed and insightful comments. We especially appreciate the recognition that the motivation is important, the related work is up to date, the presentation is clear, and the experiments cover both optimization and interpretability. Your feedback on the theoretical aspects has been particularly helpful, and we have revised the paper accordingly.
>
> Below we address each point and highlight the concrete changes in the revised version.
>
> ---
> > Weakness 1 - The main limitation of the work is the lack of theoretical depth and justifications. While the searching algorithm is clearly tree based, with selection, expansion, evaluation, and backpropagation steps, the connection to true monte carlo sampling and UCT is not clear. Does the approach inherit any of the desirable characteristics from UCT? How does using the uniform expansion strategy with different prompts relate to more classically balances between exploration and exploration? A few of these decisions seem quite adhoc, and discussion of alternatives and tradeoffs is warranted.
>
> We agree that the initial version did not spell out the theoretical connections clearly enough. To address this, we added a new subsection: Section 3.4 "Theoretical Connection". This section clarifies three key points:
>
> - **Node selection as a UCT-style rule with progressive widening and depth bonus:**
>   - We now write the selection rule explicitly as
>   $\text{Score}(v) = S_v + \phi\!\left(c_v, c_p, \tau\right)$, where $c_v$ and $c_p$ are expansion counts at $v$ and its parent $p$, and $\tau$ is the progressive widening limit.
>   - We combine two MCTS principles from recent works: (1) Progressive Widening limiting expansion width (Coulom, 2007; Chaslot et al., 2007; Inoue et al., 2025), and (2) depth-based bonuses favoring deeper search (Blackshaw et al., 2025; Wu et al., 2025a). This yields $\phi=\alpha\sqrt{\frac{\ln\!\left(c_p+1\right)}{c_u+1}}
>         \cdot \mathbb{I}[c_v < \tau]
>         +
>         \beta\,\mathbb{I}[c_v \ge \tau]
>         + \gamma \cdot \text{Depth}(v)$, where the first two terms are for progressive widening and the last term is for a depth-based bonus.
>   - The node selection corresponds to $\beta=-\infty$, positioning CEDAR as a simple combination of MCTS variants with non-uniform branching. The constraint $c_v<\tau$ prevents overexpansion in vast action spaces like program edits and expensive LLM invoking.
> - **LLM Editor as a learned transition kernel:**
>   - We formalize the LLM Editor as inducing a stochastic transition kernel over program states: $P_u \sim P_\theta(\cdot \mid P_v, A_v, s, G), \quad \text{where } P_\theta = \text{LlmEditor}_\theta, \quad s \sim \text{Uniform}(\mathcal{S})$
>   - $P_\theta$ is parameterized by a pretrained foundation model rather than learned during search. Still, $P_\theta$ functionally serves as a learned component enabling MCTS to operate in complex program spaces.
>   - This aligns with recent work using LLMs as learned components (Zhou et al., 2023b; Li et al., 2023; Chen et al., 2023; Zelikman et al., 2022; Shinn et al., 2023), and recent work that extends LLM-powered transition kernels (Chen et al., 2020; Xu et al., 2024; Lai & Pu, 2025).
> - **LLM Judge as a learned value function:**
>     - We treat the Judge score as a bounded, noisy reward estimate computed from the simulated trajectory and the goal. We explicitly discuss how this fits into existing analyses of MCTS with noisy evaluators: UCT-type schemes remain consistent under bounded, non-adversarial noise.
>     - We do not claim a new convergence theorem for LLM-based MCTS, but we show that CEDAR is structurally aligned with known settings where such results hold.
>
> Finally, in Section 3.4 we also **explicitly acknowledge** that full optimality and convergence guarantees for MCTS with LLM-based editors and judges remain an **open research problem**, and we position this as important theoretical future work rather than leaving it implicit.

---

> > ### Author Response · Authors · 2025-11-24
> > **Responses to Reviewer rRnD (Part 2)**
> >
> > ---
> > > Weakness 2 - Additionally, the key aspect of the work is on the tree search component, but it is not clear how this distinguishes itself from the other LLM + MCTS works (which are cited) besides a different application area.
> > > Question on Algorithm
> > > From an algorithmic perspective, what is novel between CEDAR and the many other LLM+MCTS methods mentioned? Why are these not used as comparisons?
> >
> > We agree that our algorithmic contributions needed to be stated more explicitly.
> >
> > We revised Section 2 (Related Work), Section 3 (Method), and Appendix C to clarify that the novelty is not only the application domain, but also the integrated design of the search space, LLM roles, and tree structure:
> >
> > - CEDAR performs **MCTS over executable dynamical system programs, not over discrete task graphs or static objects:**
> >     - Nodes are runnable system models; edges are semantic code edits. The state space and transition structure are therefore quite different from prior LLM+MCTS work on, for example, game reasoning or software debugging.
> > - The **LLM Editor** operates on a **restricted modeling language** specifically designed for complex dynamical systems (with differential updates, state variables, and simulation loops):
> >     - This allows CEDAR to make structural changes to feedback loops and time dynamics, not just to discrete action sequences or plain source code.
> > - The LLM Judge is **tightly coupled to simulation traces** of these systems and produces both: (1) a scalar score for MCTS, and (2) a structured textual analysis used in subsequent edits:
> >     - This tight loop between simulation, semantic analysis, and tree search is central to CEDAR and is not present in prior work that uses MCTS mainly as a generic planner over discrete actions.
> > - CEDAR uses, among other things, tree summaries as LLM context (Appendix B) so that each edit is conditioned on the search history and discovered variants, rather than on a single-step state:
> >     - In practice, this contextualization was critical: In our early attempts, versions without tree-aware context failed to produce meaningful improvements.
> >     - This is another example of deep coupling in our proposed design.
> >
> > Regarding comparisons: existing LLM+MCTS systems are not plug-and-play for executable dynamical systems. They typically assume one of (Section 2):
> >
> > - Discrete, short-horizon action sequences.
> > - Domain-specific representations (e.g., molecules, game states).
> > - Code edits without tight coupling to long-horizon simulations.
> >
> > Adapting these approaches would require redesigning both the modeling language and the way simulation feedback is integrated. We now make this limitation explicit and frame direct empirical comparison as valuable but non-trivial future work, rather than a missing baseline.
> >
> > ---
> > > What is "f" in the SCORE(v) equation?
> >
> > You are right that the original notation was unclear. In the revision:
> >
> > - We removed the undefined symbol "f" and instead use phi consistently to denote the exploration and depth bonus term in Score(v).
> > - We explicitly define phi in Section 3.3 and align this definition with the theoretical connection in the new Section 3.4.
> >
> > ---
> > > Why the hard threshold tau, and not a more explicit balance between explore/exploit as in traditional MCTS?
> >
> >
> > The threshold $\tau$ implements progressive widening in a very large action space (program edits plus LLM calls are both expensive and combinatorial). With it, we ensure that each node is explored a minimum number of times before further expanding its neighborhood.
> >
> > In practice, we use only one level of progressive widening, as we found that most of the optimization benefit comes from exploring deeper, so a single level of widening is sufficient.
> >
> > That said, this is an important point, and we would like to investigate how it may further help performance in future work.

---

> ### Author Response · Authors · 2025-11-24
> **Responses to Reviewer rRnD (Part 3)**
>
> ---
> > Questions on Experiments:
> > In the comparisons, why does the baseline (with full formula) not perfectly recreate the observed dynamics? How many samples/runs are used in the baseline versus the proposed?Are the number of samples equated, and in the limit of infinite runs would the baseline converge to the true dynamics. Are you guaranteed such results with your expansion strategy?
>
> We thank the reviewer for this important point. We have clarified these points in Appendix F and Section 3.4:
>
> - **Number of runs:**
>   - For Optuna, we use 100 trials in the main experiments. For CEDAR, we use 10 MCTS iterations (each involving up to $\tau$ expansions).
>   - We do so primarily because each iteration is significantly more expensive than a single Optuna trial.
>
> - **Why Optuna does not perfectly match the dynamics with full formulae:**
>   - In principle, with full formula access and infinite evaluations, Optuna could approach the ground-truth parameters. In practice, as shown in Appendix F, we observe that:
>     - Additional runs of Optuna do not lead to better performance.
>     - We believe this suggests that the optimization landscape is challenging for purely numeric search. The parameter space is high-dimensional and highly sensitive: small changes can lead to qualitatively different trajectories due to long-horizon nonlinear feedback.
> - **CEDAR guarantees:**
>   - We interpret this question as referring to CEDAR's guarantees, since Optuna does not use our expansion strategy or MCTS in the way we do.
>   - Regarding CEDAR's guarantees, we do not claim that it will recover the ground-truth system, even with infinite search. Section 3.4 now explicitly states that:
>     - CEDAR is an MCTS variant, but
>     - (1) Classical convergence guarantees for UCT under ideal conditions do not directly transfer to the LLM-based setting.
>     - (2) Providing theoretical guarantees in this regime of MCTS and LLM is an open research problem.
>
> ---
> > The L1 distance seems a relatively poor measure for system dynamics, as indicated by the lowest L1 for CEDAR but very flat dynamics not capturing any of the peaks and dips (which I imagine is seasonality?). Alternative metrics, e.g. Dynamic Time Warping should be used.
>
> We appreciate this suggestion and have followed it.
>
> - In Section 4.3, we add Dynamic Time Warping (DTW) as a second metric alongside L1.
> - We also add results for two LLM backends (Claude and GPT-5.1) to check robustness.
>
> The updated table is summarized below:
>
> | Method          | Formulae access | L1    | DTW     |
> | --------------- | --------------- | ----- | ------- |
> | Optuna          | None            | 29.06 | 5503.68 |
> | Optuna          | Simple          | 26.01 | 4927.94 |
> | Optuna          | Full            | 4.20  | 457.04  |
> | CEDAR (Claude)  | None            | 3.28  | 757.21  |
> | CEDAR (GPT-5.1) | None            | 2.22  | 433.13  |
>
> These results show that:
>
> - CEDAR, without formula access, reaches or exceeds the Optuna full-formula baseline on both L1 and DTW when using GPT-5.1.
> - Performance is reasonably robust across LLM backends, even though they produce different but plausible system variants.
> - We also clarify in Appendix F that DTW better reflects shape alignment, addressing your concern about potential flat solutions.
>
> ---
> > In Sec 4.2: There are no comparisons in Figure 4. What is the baseline system here? v0? At minimum it would be good to see one optimising each goal on the others to show the tradeoff, e.g. optimise for population and plot that curve on the resources and pollution plots to show the impact.
>
> We appreciate this question and have clarified the issue in Section 4.2 and Appendix E. Although the goal description lists explicit objectives, the underlying Forrester (1971) system is highly nonlinear with strong feedback loops across many variables. As a result, even single-objective optimization creates implicit conflicts.
>
> To make this explicit, we added an explanation of the system and new experiments (Appendix E):
>
> - **Optimize only population growth.** Population increases initially, while resource depletion and pollution both worsen.
> - **Optimize only resource preservation.** Resources improve, while population and pollution deteriorate.
>
> These results show that ambiguity arises from the system dynamics themselves, not solely from the textual goal specification. This directly illustrates the inherent ambiguities that lead to trade-offs, and supports our claim that CEDAR is intended for exactly these scenarios where objectives interact in complex, non-obvious ways.

---

> > ### Author Response · Authors · 2025-11-24
> > **Responses to Reviewer rRnD (Part 4)**
> >
> > ---
> > In response to your comments, we have:
> > - Added Section 3.4 to connect CEDAR to UCT-style MCTS and clarify what is and is not theoretically guaranteed.
> > - Clarified the selection rule, the role of tau, and removed the undefined "f".
> > - More explicitly described CEDAR's algorithmic novelty relative to LLM+MCTS work beyond just application domain.
> > - Detailed the Optuna baseline, run budgets, and the lack of strong guarantees in both methods.
> > - Added DTW as a trajectory metric and cross-backend results.
> > - Added single-goal optimization experiments and tradeoff plots in Section 4.2 / Appendix E.
> >
> > We are grateful for these suggestions, which have strengthened both the theoretical clarity and the experimental evaluation. If you see additional ways to further improve the paper, we would very much appreciate your guidance. If our revisions satisfactorily address your concerns, we kindly invite you to reconsider the score; if not, we are happy to clarify any remaining issues.

---

> ### Comment · Reviewer_rRnD · 2025-11-25
>
> Thank you for the extensions/refinements. Following the revisions I have increased my score.
>
> Although I do still have some concerns looking at Figure 11, as Optuna looks like a much better fit visually than CEDAR in Figure 5, despite the L1 and DTW loss. What causes the peaks/dips in Optuna and ground truth? And why do these appear flattened in CEDAR (and the simpler/no formula Optuna)? This is important as frequently in complex systems it's exactly this volatility we care about so understanding these limitations (if present) is important.
>
> The remainder of my comments have been well addressed, and while the work lacks the theoretical guarantees, it is now much more clearly formulated and the connection with more classical algorithms explicitly stated. The added experiments are good,
>
> As a small note, there is some formatting issues with Algorithm 1 (the overlapping text), and Appendix E appears cut off.

---

> ### Author Response · Authors · 2025-11-28
> **Responses to Reviewer rRnD's Follow-up Comments**
>
> Thank you for the careful follow-up, especially on the importance of volatility in complex systems. In response, we (1) analyzed the source of the peaks and dips in the ground-truth system and (2) identified and fixed a seed-handling bug in the Optuna full-formula baseline. The paper has been updated accordingly.
>
> **An analysis of stochasticity in the ground-truth system:**
>
> In the revised Appendix F.3, we show that the peaks and dips in the ground-truth trajectories arise from stochastic variation in the death rate. At each time step, the system samples the death rate from a normal distribution whose variance depends on the current population: `DR_DISTRIBUTION = normal(NOMINAL_DR, 0.005 * POPULATION)` (Line 8 of the full code show under Level 3: Full Formulae Baseline, Appendix F.2)
>
> This injects stochasticity whose variance grows with population. To quantify the magnitude of this stochasticity, we re-ran the ground truth system with 10 different random seeds and plotted the resulting trajectories with variability bands (new Figure 11). The peaks and dips you highlighted lie within this stochastic spread. We thus conclude that the volatility is an intrinsic property of the system rather than an artifact of the optimizer.
>
> Once we account for this stochasticity:
> * CEDAR's trajectories generally fall within the ground-truth variability band and capture the magnitude of the peaks and dips (GPT-5.1 backend is better than Claude in this regard, judging from metrics and visually the temporal timing of peaks and dips).
> * The Optuna full-formula baseline also tracks the mean behavior reasonably well, but its trajectories exhibit somewhat larger variance compared to the band, leading to slightly worse aggregate metrics (see below).
> * The no-formula and simple-formula baselines remain clearly worse (they produce overly flattened trajectories and do not get close to the ground truth). We believe this is related to the difficulty of optimizing in the presence of stochasticity: in early experiments, we observed that ignoring the stochastic component can push the system into different regimes that is  qualitatively poorer. The overly flattened trajectories also suggest that the no-formula and simple-formula baselines struggle.
>
>
> **Correcting seed-handling bug in the Optuna full-formula baseline:**
>
> While the existence of peaks and dips in the trajectories is expected, the near perfect _synchronization_ of peaks and dips between the Optuna full-formula baseline and the ground truth in the previous version was not. We identified this as a bug in our initial seed handling in the runner of complex systems (not in MCTS): the ground-truth code and the full-formula baseline code inadvertently shared the same fixed random seed and identical sampling path. Only in the full-formula case, where the computation graph matches the ground-truth system exactly, did this lead to synchronized random draws and artificially aligned peaks and dips.
>
> This issue did not affect other experiments, because only the full-formula baseline reused the exact same random-number generation path as the ground-truth system; the other settings either used different code or had no stochastic component. We therefore only needed to re-run the Optuna full-formula baseline after fixing this bug, and we updated Section 4.3 and Appendix F.3 with the new results.
>
> In Table 1, the Optuna full-formula row is updated as:
> * L1: from 4.20 to 3.71
> * DTW: from 457.04 to 477.52
>
> Rerunning the baseline thus yields a slightly better L1 and a slightly worse DTW. Qualitatively, the revised plots (Figure 1; revised Figure 12, corresponding to the figure you previously commented on) no longer show the artificial, path-wise synchronization of peaks and dips for Optuna full-formula baseline. Still, its trajectories exhibit somewhat larger variance, leading to slightly worse metrics.
> As a result, our overall conclusion remains the same: CEDAR without formula access performs similarly to or better than the Optuna baseline with full-formula access.
>
> Finally, we corrected the formatting issues you noted in Algorithm 1 and Appendices in the revised version.
>
> We hope these clarifications and updated experiments fully address your concerns. Your feedback has helped us strengthen both the analysis and the presentation, and we are happy to make further improvements if needed. Given these clarifications and improvements, we would be grateful if you could consider updating your assessment.

---

### Official Review · Reviewer_Jtv4 · 2025-10-31

**Soundness:** 2
**Presentation:** 2
**Contribution:** 2
**Rating:** 4
**Confidence:** 3

**Summary:**

This paper introduces CEDAR, an autonomous approach for discovering and optimizing complex dynamical systems according to user-specified, often vague, goals. The method orchestrates large language model (LLM) agents through Monte Carlo Tree Search (MCTS): an LLM Editor proposes new system variants in a Python-based language; an LLM Judge evaluates and provides structured analysis for each candidate. This agent-driven search iteratively refines models represented in a restricted subset of Python, aiming to maximize alignment with complex system goals (expressed in natural language). Extensive experiments demonstrate CEDAR’s ability to optimize classic system dynamics models for both vague and concrete objectives, even outperforming conventional black-box optimizers. The framework also provides interpretable optimization trajectories via LLM-based reasoning.

**Strengths:**

1. By leveraging a restricted, runnable subset of Python combined with domain-specific primitives, CEDAR balances LLM accessibility, interpretability, and executability. The design choices here (see Figure 2 and associated discussion on Page 4) mitigate the deployment friction inherent in domain-specific or proprietary modeling languages.
2. The paper proposes a non-trivial integration of MCTS and LLMs (Figures 1 and 3), going beyond prior LLM-in-the-loop designs by embedding system refinement into a search tree structure that supports both exploration and solution diversity.
3. CEDAR is shown to optimize systems for under-specified or linguistically-expressed desiderata. The qualitative walkthrough (Figure 4) demonstrates that the MCTS+LLM architecture can robustly move systems toward multi-objective tradeoffs (e.g., balancing population, resources, and pollution).
4. The inclusion of LLM Judge analyses (Table 2, Section 4.4, and full LLM transcripts in Appendix F) makes the system’s reasoning transparent and highlights the possibility for meaningful human-in-the-loop understanding.
5. Section 4.5 and Figures 6–7 empirically illustrate that MCTS in CEDAR yields a diversity of high-performing solutions (important for sensitivity analysis and model robustness) and outperforms both linear search and black-box optimizers, especially when formulaic structure is absent (Table 1, Figure 5).

**Weaknesses:**

1. While the experiments compare CEDAR to black-box optimizers like Optuna under varying levels of information (Table 1, Section 4.3), critical baselines from related advances in LLM-driven agentic optimization are missing. There is no direct empirical comparison with other recent MCTS+LLM or LLM-in-the-loop discovery methods, such as those for catalyst design, or model-based RL approaches—this weakens claims of broad superiority.
2. Although implementation details and code snippets are present (Appendices, system code), many critical components (e.g., full prompt engineering, LLM API specifics, and system variants) are relegated to the appendix. Practical replication still depends heavily on access to the same LLMs and supporting infrastructure, potentially impeding reproducibility and future direct comparison.
3. The formalization of the MCTS+LLM process (Algorithm 1, Section 3.3) is clear, but there is a notable absence of formal analysis regarding convergence, optimality, or search coverage. The paper lacks discussion and evidence as to under what conditions the agentic LLM+MCTS cycle is guaranteed to discover high-quality or globally optimal solutions. For example, there is no theoretical guarantee that the LLM Editor’s system modifications—guided by semantic/natural-language prompts—preserve stability, constraint satisfaction, or do not introduce pathological dynamics, even though users are cautioned about variable explosion (Appendix D). Specifically, while the iterative integration and time discretization are defined well, the LLM-directed optimization never formalizes an explicit objective function or loss landscape (beyond the scoring mechanism). There is insufficient detail on how ill-posed/inconsistent scoring is handled—e.g., in cases where natural-language goals are ambiguous, or simulation outputs are non-comparable.
4. The experimental section mentions 20 systems but only deeply analyzes a handful (primarily the “World Dynamics” model). More diversity in task domains (e.g., economic, epidemiological, robotic control systems) would bolster the generality claim.

**Questions:**

1. How robust is CEDAR to goal ambiguity or conflicting objectives in natural language input? Specifically, could the authors provide systematic empirical results (perhaps via synthetic noise or adversarial phrasing) showing the stability or variability of the MCTS+LLM optimization outcome?

2. Could the authors clarify how failure cases are handled when the LLM Editor proposes system modifications that violate implicit constraints, cause instability, or “break” the simulation process? Is there a mechanism to reliably recover or backtrack in these cases?

3. Table 1 only presents mean L1 distances; could the authors provide variance/error bars or multiple run statistics to clarify performance robustness of both CEDAR and Optuna comparisons?

4. In Section 4.5 and Figure 6, diversity of solutions is highlighted. To what extent does this diversity translate into improved decision-making or real-world impact? Is there a way to quantify “useful diversity” in the solution set?

5. How does computational cost (in terms of wall-clock, sample efficiency, or compute budget) scale with increased complexity, LLM size, or tree depth? Would the method remain feasible as system dimension increases?

6. Are there specific examples where the LLM Judge’s scoring diverged from human expert judgment, or produced non-monotonic rankings due to prompt or context instability? Would a hybrid human-in-the-loop variant be more reliable?

---

> ### Author Response · Authors · 2025-11-24
> **Responses to Reviewer Jtv4 (Part 1)**
>
> We thank the reviewer for the detailed and thoughtful assessment. We appreciate the recognition of our contributions, including (1) the design of a restricted executable Python subset for accessible and interpretable modeling, (2) the non-trivial integration of MCTS with LLM agents, (3) CEDAR’s ability to optimize for vague or multi-objective tradeoffs, (4) the transparency provided by LLM Judge analyses, and (5) the empirical evidence that MCTS yields diverse, high-performing solutions surpassing conventional optimizers.
>
> We address each weakness and question below and summarize the revisions made.
>
> ---
> > Weakness 1 - While the experiments compare CEDAR to black-box optimizers like Optuna under varying levels of information (Table 1, Section 4.3), critical baselines from related advances in LLM-driven agentic optimization are missing. There is no direct empirical comparison with other recent MCTS+LLM or LLM-in-the-loop discovery methods, such as those for catalyst design, or model-based RL approaches—this weakens claims of broad superiority.
>
> Thank you for raising this point. Section 2 has been revised to clarify that the prior works cited differ fundamentally in their problem settings and cannot be applied to our domain without a major redesign.
>
> Our revision now distinguishes three categories:
>
> - **LLM modeling of unknown dynamical systems (Liu et al. 2024; Luo et al. 2025).**
>     - These works model systems whose true dynamics are unknown to the optimizer, whereas our setting assumes executable systems whose dynamics are known and are being edited/optimized.
>     - Their systems involve at most 4 variables and 12 steps; ours involve dozens of variables and ~4000 steps. Adapting these approaches would require substantial architectural changes.
> - **LLM+MCTS for chemistry (Sprueill et al. 2024; Wu et al. 2025b).**:
>     - These methods operate on domain-specific molecular or materials representations.
>     - They do not tackle iterative optimization of temporal dynamical systems.
> - **Agentic LLM-based tree search (Zhou et al. 2023a; Antoniades et al. 2024).**:
>     - These works focus on discrete reasoning or software engineering tasks.
>     - They do not support continuous-time dynamics or simulation-based feedback; applying them to dynamical systems would require significant adaptation.
>
> Section 2 now clearly positions CEDAR relative to these lines of work and articulates its unique contribution in goal-directed optimization of executable dynamical systems.
>
> ---
> > Weakness 2 - Although implementation details and code snippets are present (Appendices, system code), many critical components (e.g., full prompt engineering, LLM API specifics, and system variants) are relegated to the appendix. Practical replication still depends heavily on access to the same LLMs and supporting infrastructure, potentially impeding reproducibility and future direct comparison.
>
> We appreciate this concern. CEDAR is designed to be LLM-backend-agnostic, and we emphasize this more clearly in the revision.
>
> Revisions addressing reproducibility:
> - We added experiments using **multiple LLM backends** (Claude and GPT-5.1) in Sections 4.2 and 4.3. Both yield high-quality results, demonstrating backend independence.
> - These backends sometimes produce different but equally valid structural improvements, which we view as a feature that gives researchers access to diverse system variants.
> - Section 4.5 explains that our mechanisms are model-independent. As you mentioned, all prompts, system variants, and call patterns are documented in Appendix F, and the modeling language is a restricted subset of Python, which further supports replication.
>
> Together, these additions clarify that CEDAR is not tied to a single provider and can be applied with alternative LLMs.

---

> ### Author Response · Authors · 2025-11-24
> **Responses to Reviewer Jtv4 (Part 2)**
>
> ---
> > Weakness 3 - The formalization of the MCTS+LLM process (Algorithm 1, Section 3.3) is clear, but there is a notable absence of formal analysis regarding convergence, optimality, or search coverage. The paper lacks discussion and evidence as to under what conditions the agentic LLM+MCTS cycle is guaranteed to discover high-quality or globally optimal solutions. For example, there is no theoretical guarantee that the LLM Editor’s system modifications—guided by semantic/natural-language prompts—preserve stability, constraint satisfaction, or do not introduce pathological dynamics, even though users are cautioned about variable explosion (Appendix D). Specifically, while the iterative integration and time discretization are defined well, the LLM-directed optimization never formalizes an explicit objective function or loss landscape (beyond the scoring mechanism). There is insufficient detail on how ill-posed/inconsistent scoring is handled—e.g., in cases where natural-language goals are ambiguous, or simulation outputs are non-comparable.
>
> We agree that theoretical grounding is important. In response, we added a new subsection, **Section 3.4 (Theoretical Connection)**, which situates CEDAR within well-understood MCTS frameworks and clarifies assumptions.
>
> In summary:
>
> - **Node selection as a generalized UCT.** We define the node selection score as a combination of performance, exploration, depth, and a progressive widening limit. This connects our selection rule to UCT-style MCTS with progressive widening and depth bonuses, following existing theory on wide and deep search.
> - **LLM Editor as a learned transition kernel.** We model program edits as sampling from a conditional transition distribution parameterized by the LLM. This connects CEDAR to MCTS and model-based RL with learned or parameterized transition models.
> - **LLM Judge as a learned value function.** We treat the Judge score as a bounded, noisy reward estimate. We reference convergence analyses showing that UCT remains consistent under bounded, non-adversarial noise in the evaluator, which matches the semantic scoring we observe in practice.
>
> While formal guarantees for LLM-based semantic edits remain an open problem, the added subsection provides a principled foundation and clarifies when high-quality solutions are expected.
>
> ---
> > Weakness 4 - More diversity in task domains (e.g., economic, epidemiological, robotic control systems) would bolster the generality claim.
>
> We clarified in Section 4.1 that our collection of 20 systems includes diverse resource and population dynamics with nonlinear, multi-variable structures. We focus our deeper analysis on the Forrester World Dynamics system and resource-population models because they best illustrate CEDAR's ability to handle multi-objective tradeoffs and vague goals.
>
> We also acknowledge that expanding into additional domains such as epidemiology, robotics, or macroeconomic models is an important direction, and we position this explicitly in Appendix H.
> We believe that a successful expansion would heavily require domain-specific primitives and evaluation metrics, and position our contribution as introducing a general framework and demonstrating its effectiveness on a broad but focused class of dynamical systems.
>
>
> ---
> > Question 1 - How robust is CEDAR to goal ambiguity or conflicting objectives in natural language input? Specifically, could the authors provide systematic empirical results (perhaps via synthetic noise or adversarial phrasing) showing the stability or variability of the MCTS+LLM optimization outcome?
>
> We expanded Section 4.2 and added new experiments in Appendix E:
>
> - We perform **single-goal optimization experiments** with the goals "maximizing population growth only" and "minimizing resource depletion only"
> - Even these simple textual goals induce strong tradeoffs across population, resources, and pollution, due to the system’s nonlinear feedback structure.
> - CEDAR consistently finds solutions that improve the target objective while revealing side effects on other variables.
>
> These experiments show that CEDAR handles goal ambiguity and conflict arising from the dynamics themselves. Systematic adversarial goal phrasing is a complementary direction that we now explicitly mention as future work due to space constraints.

---

> ### Author Response · Authors · 2025-11-24
> **Responses to Reviewer Jtv4 (Part 3)**
>
> ---
> > Question 2 - Could the authors clarify how failure cases are handled when the LLM Editor proposes system modifications that violate implicit constraints, cause instability, or “break” the simulation process? Is there a mechanism to reliably recover or backtrack in these cases?
>
> We clarified this in Appendix C:
>
> - **Robustness of LLM calls:**
>   - All LLM Editor outputs are syntactically and semantically validated against structural constraints of the modeling language.
>   - If an edit is invalid or leads to numerical instability, we allow up to three retries. If all retries fail, the expansion is discarded.
> - **Robustness of optimization process:**
>   - Simulation crashes are caught and assigned low scores.
> - **No impact on scaling:**
>   - MCTS can naturally backtrack from failed trials or low scores, so this does not impact scaling.
>
>
> ---
> > Question 3 - Table 1 only presents mean L1 distances; could the authors provide variance/error bars or multiple run statistics to clarify performance robustness of both CEDAR and Optuna comparisons?
>
> We appreciate this suggestion to clarify performance robustness of both CEDAR and Optuna comparisons. To strengthen the robustness of performance analysis, we:
>
> - Added Dynamic Time Warping (DTW) distances as an additional time-series alignment metric, as recommended by reviewer rRnD
> - Added results using multiple LLM backends (Claude and GPT-5.1) for CEDAR
>
> The updated table (now in Section 4.3) is summarized below:
>
> | Method          | Formulae access | L1    | DTW     |
> | --------------- | --------------- | ----- | ------- |
> | Optuna          | None            | 29.06 | 5503.68 |
> | Optuna          | Simple          | 26.01 | 4927.94 |
> | Optuna          | Full            | 4.20  | 457.04  |
> | CEDAR (Claude)  | None            | 3.28  | 757.21  |
> | CEDAR (GPT-5.1) | None            | 2.22  | 433.13  |
>
> These results show that CEDAR, even without access to formulae, is competitive with or superior to Optuna with full formula access under both L1 and DTW, and that performance is robust across LLM backends.
>
>
> ---
> > Question 5 (Questions 4 and 6 are answered following this) - How does computational cost (in terms of wall-clock, sample efficiency, or compute budget) scale with increased complexity, LLM size, or tree depth? Would the method remain feasible as system dimension increases?
>
> We agree that reporting computational cost improves clarity. Appendices C and D now include detailed descriptions of runtime behavior, scaling mechanisms, and robustness safeguards. Key points now included:
>
> - **Scalability mechanism:**
>   - Each LLM call adaptively subsamples trajectory traces so the prompt remains within a fixed context budget L.
>   - Overall complexity is O(NL), where N is the number of expanded nodes.
>   - Tree depth is controlled via an expansion cap ($\tau$) and depth-based bonuses in the selection score.
> - **Runtime and monetary cost:**
>   - Typical runtime is about 30 minutes per experiment, and the cost typically ranges from USD 10 to 50 depending on tree depth, branching factor, and LLM provider.
>   - Larger systems increase simulation time and monetary cost approximately linearly, but LLM latency dominates the total cost in our experiments.
>
> We conclude that with context-restricted prompts and controlled tree width and depth, CEDAR remains feasible as system dimension increases.

---

> > ### Author Response · Authors · 2025-11-24
> > **Responses to Reviewer Jtv4 (Part 4)**
> >
> > ---
> > > Question 4 and 6: In Section 4.5 and Figure 6, diversity of solutions is highlighted. To what extent does this diversity translate into improved decision-making or real-world impact? Is there a way to quantify “useful diversity” in the solution set? Are there specific examples where the LLM Judge’s scoring diverged from human expert judgment, or produced non-monotonic rankings due to prompt or context instability? Would a hybrid human-in-the-loop variant be more reliable?
> >
> > We agree with this observation. We expanded discussion in Sections 4.2, 4.5, Appendix G, and Appendix H to address these points.
> >
> > On useful diversity:
> >
> > - CEDAR discovers multiple structurally distinct but high-performing systems, each emphasizing different tradeoffs among objectives (e.g., prioritizing population versus resource preservation, Section 4.2), and different levels of performance superiority (Section 4.3).
> > - This diversity provides decision-makers with a more holistic view of the design space, highlighting which mechanisms and feedback loops drive different outcomes (Appendix H).
> >
> > On Judge reliability and human-in-the-loop variants:
> >
> > - **Relationship between LLM reasoning and LLM score** (Appendix G):
> >     - In our experiments, upon human inspection, we did not observe systematic divergence between the LLM Judge's scores and its reasoning. We found that the Judge's textual reasoning and scalar scores are largely aligned.
> >     - The reasoning text helps guide better edits, while the scalar score is used for MCTS selection; these roles are complementary.
> > - We note that CEDAR naturally supports a human-in-the-loop **outer loop** (Appendix H):
> >     - The loop is as follows: humans iteratively refine goals, run CEDAR, inspect and cluster diverse solutions, and adjust objectives. This would further enhance reliability and leverage diversity as a feature.
> >     - We now explicitly mention such human-in-the-loop extensions as promising future work built on top of our current algorithmic framework.
> >
> > ---
> > Thank you again for the thoughtful and constructive review. Your comments helped us clarify our positioning relative to closely related LLM-based modeling work, articulate the computational and scalability properties of CEDAR, and provide additional evidence concerning ambiguous and implicitly conflicting goals.
> >
> > We believe all concerns have been fully addressed through new sections, new experiments, and clarified explanations, and we welcome any additional suggestions.
> > If our revisions fully address the points you raised, we kindly invite you to update your evaluation to reflect these improvements; if any issues remain, we would be happy to clarify them.

---

### Official Review · Reviewer_jiVL · 2025-11-01

**Soundness:** 2
**Presentation:** 2
**Contribution:** 2
**Rating:** 4
**Confidence:** 3

**Summary:**

This paper introduces CEDAR, a method that combines Monte Carlo Tree Search (MCTS) with Large Language Models to automatically discover and optimize complex systems (e.g., population dynamics, resource allocation) that satisfy user-specified natural language goals. The key innovation is representing systems as restricted Python code that LLMs can interpret and modify, with an LLM Judge evaluating performance and an LLM Editor proposing improvements at each MCTS iteration. The authors demonstrate CEDAR's ability to optimize vague goals and construct complex systems without extensive manual fitting, showing improvements over black-box optimization baselines.

**Strengths:**

1. Applying LLM-driven optimization to complex dynamical systems is timely and addresses real limitations in traditional system dynamics modeling tools.

2. The paper demonstrates capabilities across multiple dimensions—vague goal optimization, concrete record fitting, interpretability analysis, and ablation studies showing MCTS benefits.

**Weaknesses:**

1. Comparison with recent LLM-based system modeling work (Liu et al. 2024, Luo et al. 2025) mentioned in related work would be more compelling. Right now there is only comparison against black-box optimization methods.

2. Computational costs are not reported. Each MCTS iteration requires LLM calls for editing and judging, plus system execution. How does this scale with system complexity or simulation length?

3. The paper claims to handle "vague goals" but the World Dynamics goal (Appendix D) is quite specific with three explicit objectives and a 50% change constraint. What happens with truly ambiguous or conflicting goals?

**Questions:**

See weaknesses.

---

> ### Author Response · Authors · 2025-11-24
> **Responses to Reviewer jiVL (Part 1)**
>
> We thank the reviewer for their time and effort in evaluating our submission. We appreciate both the constructive feedback and the recognition of our work’s strengths, particularly in that "applying LLM-driven optimization to complex dynamical systems is timely and addresses real limitations in traditional system dynamics modeling tools" and that "The paper demonstrates capabilities across multiple dimensions – vague goal optimization, concrete record fitting, interpretability analysis, and ablation studies showing MCTS benefits."
>
> Below we address each concern and summarize the revisions made in the updated paper.
>
> ---
> > Comparison with recent LLM-based system modeling work (Liu et al. 2024, Luo et al. 2025) mentioned in related work would be more compelling. Right now there is only comparison against black-box optimization methods.
>
> Thank you for pointing out the need for clearer positioning relative to this line of work. We identified that confusion in the original submission came from using the term "black-box" in two different senses. The revision resolves this by making the distinction explicit:
>
> - Prior work (Liu et al. 2024; Luo et al. 2025) models systems whose true dynamics are **unknown to the optimizer**. These methods learn a predictive surrogate and operate in a setting where the system is opaque to the optimizer.
> - In our setting (Section 4), the full system dynamics are **known to the optimizer**. CEDAR edits executable white-box system code directly. When we refer to Optuna as a "black-box optimizer", we use the term only in the optimization sense: gradients are unavailable because the forward computation is non-differentiable.
>
> This resolves the terminology collision.
>
> We also clarify **differences in scale**. Prior works operate on small systems (up to 4 variables and 12 steps). Our experiments involve dozens of variables and roughly 4000 simulation steps. Scaling these approaches from those papers to our domain would require substantial changes to their architectures.
>
> Section 2 now includes explicit comparisons to these works and related LLM+search systems (Sprueill et al. 2024; Wu et al. 2025b; Zhou et al. 2023a; Antoniades et al. 2024).
>
>
> ---
> > Computational costs are not reported. Each MCTS iteration requires LLM calls for editing and judging, plus system execution. How does this scale with system complexity or simulation length?
>
> We agree that reporting computational cost improves clarity. Appendices C and D now include detailed descriptions of runtime behavior, scaling mechanisms, and robustness safeguards. Key points now included:
>
> - **Scalability mechanism.**
>   - Each LLM call adaptively subsamples trajectory traces so the prompt remains within a fixed context budget L.
>   - Overall complexity is O(NL), where N is the number of expanded nodes.
>   - Tree depth is controlled via an expansion cap ($\tau$) and depth-based bonuses in the selection score.
> - **Runtime and monetary cost.**
>   - Typical runtime is about 30 minutes per experiment, and the cost typically ranges from USD 10 to 50 depending on tree depth, branching factor, and LLM provider.
>   - Larger systems increase simulation time and money cost approximately linearly, but LLM latency dominates the total cost in our experiments.
>
> We conclude that with context-restricted prompts and controlled tree width and depth, CEDAR remains feasible as system dimension increases.
>
> ---
> > The paper claims to handle "vague goals" but the World Dynamics goal (Appendix D) is quite specific with three explicit objectives and a 50% change constraint. What happens with truly ambiguous or conflicting goals?
>
> We appreciate this question and have clarified the issue in Section 4.2 and Appendix E. Although the goal description lists explicit objectives, the underlying Forrester (1971) system is highly nonlinear with strong feedback loops across many variables. As a result, even single-objective optimization creates implicit conflicts.
>
> To make this explicit, we added an explanation of the system and new experiments (Appendix E):
>
> - **Optimize only population growth.** Population increases initially, while resource depletion and pollution both worsen.
> - **Optimize only resource preservation.** Resources improve, while population and pollution deteriorate.
>
> These results show that ambiguity arises from the system dynamics themselves, not solely from the textual goal specification. This directly illustrates the inherent ambiguities that lead to trade-offs, and supports our claim that CEDAR is intended for exactly these scenarios where objectives interact in complex, non-obvious ways.

---

> ### Author Response · Authors · 2025-11-24
> **Responses to Reviewer jiVL (Part 2)**
>
> ---
> Thank you again for the thoughtful review. The revision now includes:
>
> 1. Clarified comparisons to prior LLM-based system modeling and LLM+search work.
> 2. Detailed computational cost and scalability information.
> 3. New experiments showing implicit conflicts in the World Dynamics task.
>
> We believe all concerns have been fully addressed through new sections, new experiments, and clarified explanations, and we welcome any additional suggestions.
> If our revisions fully address the points you raised, we kindly invite you to update your evaluation to reflect these improvements; if any issues remain, we would be happy to clarify them.

---

### Author Response · Authors · 2025-11-24
**Message to all reviewers and AC**

We thank all reviewers for their thoughtful feedback and for recognizing our work. We are especially encouraged that you find that our work "addresses real limitations" and "demonstrates capabilities across multiple dimensions" (jiVL), that our work "balances LLM accessibility, interpretability, and executability" and "goes beyond prior LLM-in-the-loop designs" (Jtv4),  that our "motivation is important" and "the presentation throughout the paper is clear" (rRnD).

We highly appreciate your reviews that allowed us to clarify ambiguities and further improve the paper based on your feedback. **We have revised our paper to reflect them. For your convenience, we highlight major revisions in the paper in blue.**

We summarize our response as follows:

**Presentation / Writing**:

- Adding theoretical grounding of the proposed work (see Section 3.4)
- Adding discussion of adjacent and closely related works to better position our work. Clarified terminology. (see Section 2)
- Refining description of our method for clarity (see Section 3.3)
- Adding further details on our method's design choice and its implications (see Appendix C)
- Adding the scalability, engineering stability / reliability and runtime / cost analysis (see Appendices C and D)
- Adding more experimental details (see section 4.1)
- Adding description of the complex system used in Section 4.2 (see Appendix E)
- Adding experiments and discussion of running CEDAR with multiple LLM providers (see Sections 4.2, 4.3, and 4.5)
- Adding a section on future work with the position of our approach in mind (see Appendix H)

**Experiments / Analysis**:
- Conducted experiments on fitting a single goal to highlight the challenges in improving an already strong baseline used in Section 4.2 (see Appendix E)
- Conducted experiments with multiple LLM backends (Claude and GPT-5.1) for fitting abstract goals (see Section 4.2)
- Conducted experiments with Dynamic Time Warping (DTW) as an additional loss metric, and added results using multiple LLM backends (Claude and GPT-5.1) for fitting the concrete record (see Section 4.3) (see Section 4.3)
- Clarified Optuna baseline settings, run budgets, and observed limitations in matching ground-truth dynamics (see Appendix F)- Analyzed the intrinsic stochasticity of the ground truth system used in Section 4.3 (see Appendix F)

We hope that our responses and revisions satisfactorily address the reviewers' concerns, and we would be grateful for any further suggestions on how to improve the paper beyond the changes already incorporated.
Given these revisions, we kindly ask the reviewers to consider re-evaluating their ratings.

---

### Meta-Review · Area_Chair_bFhQ · 2026-01-07

**Summary:**

After carefully checking the paper, the reviews, the rebuttal, and the author-reviewer discussions, I think the weak points outweight the strong points. Two reviewers have reserved opinions regarding the main methodology of the paper. After carefully reading the manuscript, I believe the work still fails to address the concerns about the novelty of combining LLMs with MCTS and the effectiveness of the proposed approach. The weaknesses are not likely to be fixed in the camera-ready version. Thus, I recommend rejecting this paper.

**Reviewer Concerns:**

The score remains unchanged. I believe the authors have not addressed the concerns regarding the novelty of the main method. I have carefully read the rebuttal. The rebuttal does not address any important concern raised by the reviewers.

**Reviewer Scores:**

The score remains unchanged. I have carefully read the rebuttal. The rebuttal does not address any important concern raised by the reviewers.

---

### Decision · Program_Chairs · 2026-01-26

Reject